# Loss of Fam60a, a Sin3a subunit, results in embryonic lethality and is associated with aberrant methylation at a subset of gene promoters

Ryo Nabeshima[1,2], Osamu Nishimura[3,4], Takako Maeda[1], Natsumi Shimizu[2], Takahiro Ide[2], Kenta Yashiro[1†], Yasuo Sakai[1], Chikara Meno[1], Mitsutaka Kadota[3,4], Hidetaka Shiratori[1†], Shigehiro Kuraku[3,4]*, Hiroshi Hamada[1,2]*

[1]Developmental Genetics Group, Graduate School of Frontier Biosciences, Osaka University, Suita, Japan; [2]Laboratory for Organismal Patterning, RIKEN Center for Developmental Biology, Kobe, Japan; [3]Phyloinformatics Unit, RIKEN Center for Life Science Technologies, Kobe, Japan; [4]Laboratory for Phyloinformatics, RIKEN Center for Biosystems Dynamics Research, Kobe, Japan

**\*For correspondence:**
shigehiro.kuraku@riken.jp (SK);
hiroshi.hamada@riken.jp (HH)

†These authors contributed equally to this work

**Abstract** We have examined the role of *Fam60a*, a gene highly expressed in embryonic stem cells, in mouse development. Fam60a interacts with components of the Sin3a-Hdac transcriptional corepressor complex, and most *Fam60a*$^{-/-}$ embryos manifest hypoplasia of visceral organs and die in utero. Fam60a is recruited to the promoter regions of a subset of genes, with the expression of these genes being either up- or down-regulated in *Fam60a*$^{-/-}$ embryos. The DNA methylation level of the Fam60a target gene *Adhfe1* is maintained at embryonic day (E) 7.5 but markedly reduced at E9.5 in *Fam60a*$^{-/-}$ embryos, suggesting that DNA demethylation is enhanced in the mutant. Examination of genome-wide DNA methylation identified several differentially methylated regions, which were preferentially hypomethylated, in *Fam60a*$^{-/-}$ embryos. Our data suggest that Fam60a is required for proper embryogenesis, at least in part as a result of its regulation of DNA methylation at specific gene promoters.
DOI: https://doi.org/10.7554/eLife.36435.001

## Introduction

The Sin3a protein is a core component of a mammalian transcriptional corepressor complex that includes histone deacetylases (Hdacs) (*Hassig et al., 1997*; *Zhang et al., 1997*). Although Sin3a does not bind DNA on its own, it provides a scaffold for several transcription factors with specific DNA binding activities and thereby promotes the recruitment of Hdacs to and consequent repression of specific target genes. Mice lacking Sin3a die during embryogenesis around the time of implantation (*McDonel et al., 2012*), suggesting that the Sin3a-Hdac complex is essential for early embryonic development. Although this complex was initially thought only to repress gene expression, it can also stimulate transcription in a manner dependent on cellular context (*Icardi et al., 2012*).

Gene expression is also regulated by DNA methylation. In mammals, DNA methylation occurs predominantly at CpG sequences, with ~70% of gene promoters in mammalian genomes containing CpG islands. In general, CpG islands of transcriptionally active promoters are not methylated, whereas methylation of CpG in a promoter is associated with transcriptional silencing. During mouse development, the methylation pattern of genomic DNA is established at the peri-implantation stage

**eLife digest** As an embryo develops, its cells continue to divide and transform from unspecialized embryonic stem cells into the specialized cells that form the tissues and organs of the adult body. This complex process is controlled by a network of genes. Although most adult cells carry the same genes, different cell types each activate specific sets of genes, which ultimately gives them their unique properties. Likewise, developing cells also have unique patterns of gene expression that guide the cell's development, behavior and its interaction with neighboring cells.

For example, the gene Fam60a is highly active in embryonic stem cells, but until now, it was not known what role this gene had. To investigate this further, Nabeshima et al. studied mice that either had normal levels of Fam60a or reduced levels of Fam60a. The results showed that at a normal level, Fam60a was responsible for the intestines to develop properly. The guts of mice with reduced levels, however, grew very slowly.

Moreover, Farm60a appears to regulate several other genes, and their activity was no longer controlled properly in these mice. Nabeshima et al. discovered that this was because Fam60a could interact with protein complexes responsible for repressing or activating genes. By changing the activity of these complexes, Fam60a could affect the activity of many other genes.

A next step will be to find out how exactly Fam60a interacts with the protein complexes that affect the activity of genes. A better knowledge of how genes contribute to the development of an embryo may help understand the causes of miscarriage and find ways to prevent it.
DOI: https://doi.org/10.7554/eLife.36435.002

by the de novo methyltransferases Dnmt3a and Dnmt3b. Once established, this methylation pattern is faithfully maintained by Dnmt1 during DNA replication. The precise formation and maintenance of the DNA methylation pattern are essential for mouse embryogenesis, given that embryos lacking Dnmt enzymes develop pronounced morphological defects and die in utero (*Li et al., 1992*; *Okano et al., 1999*).

Methylated DNA can undergo demethylation, a process mediated by the Tet family of 5-methyl-cytosine dioxygenases that catalyze the conversion of 5-methylcytosine (5mC) to 5-hydroxymethylcy-tosine (5hmC) (*Tahiliani et al., 2009*). Demethylation of DNA by Tet proteins serves to activate gene promoters, but these proteins are also able to regulate gene expression via histone modification (*Wu et al., 2011*; *Wu and Zhang, 2017*). Strict regulation of Tet proteins is also required for proper development, given that mouse embryos lacking Tet1 and Tet2 as well as chimeric embryos that include cells deficient in Tet1, Tet2, and Tet3 become malformed (*Dawlaty et al., 2014*; *2013*). The mechanisms responsible for such Tet regulation have remained unknown, however.

We have previously identified *Fam60a* (*Sinhcaf*) as a gene of unknown function (gene 226 reported in [*Saijoh et al., 1996*]) that is highly expressed in mouse embryonic stem (ES) cells and whose expression in these cells is down-regulated on their differentiation. We have now examined the role of Fam60a in mouse development. Our data show that Fam60a is an embryonic component of the Sin3a-Hdac corepressor complex and regulates gene expression at least in part by regulating DNA methylation at a subset of gene promoters.

## Results

### Fam60a interacts with components of the Sin3a-Hdac complex

To examine the biochemical function of Fam60a, we generated mice harboring a *Fam60a::Venus* BAC (bacterial artificial chromosome) transgene (*Figure 1—figure supplement 1A*; as described below, the Fam60a-Venus fusion protein encoded by this transgene is functional). Immunostaining revealed that the Fam60a-Venus protein was present in nuclei of embryonic day (E) 9.5 embryos harboring the transgene, and that Fam60a was localized to the nucleus of undifferentiated P19 (mouse embryonic carcinoma) cells (*Figure 1—figure supplement 2*).

To identify proteins that might interact with Fam60a, we prepared nuclear extracts from E10.5 embryos harboring the *Fam60a::Venus* transgene under three different conditions, subjected the extracts to immunoprecipitation with antibodies to green fluorescent protein (GFP), and analyzed

the precipitated proteins by mass spectrometry. The major proteins identified were Arid4a, Arid4b, Sin3a, Sap130, Hdac1, Hdac2, Suds3, and Brms1l (*Figure 1A*), all of which are components of the Sin3a-Hdac corepressor complex (*Cunliffe, 2008*; *Fleischer et al., 2003*; *Grzenda et al., 2009*; *Nikolaev et al., 2004*; *Shiio et al., 2006*; *Silverstein and Ekwall, 2005*). Arid4a and Arid4b were not detected if nuclear extracts were prepared with radioimmunoprecipitation assay (RIPA) buffer (*Figure 1A*), the most stringent of the three conditions used, suggesting that these proteins interact weakly with the other components of the Sin3a-Hdac complex (*Lai et al., 2001*).

Further co-immunoprecipitation analysis confirmed that Fam60a interacts with components of the Sin3a-Hdac complex. Immunoprecipitates prepared from nuclear extracts of E10.5 wild-type (WT) embryos with antibodies to Fam60a were thus found to contain Sin3a, Hdac1, and Hdac2 (*Figure 1B*). In addition, these three proteins were detected in immunoprecipitates prepared from nuclear extracts of *Fam60a::Venus* transgenic embryos with antibodies to GFP (*Figure 1B*). Immuno-precipitates prepared from undifferentiated P19 cells with antibodies to Fam60a also contained Sin3a and Hdac1 but not Hdac2 (*Figure 1B*). Reciprocal co-immunoprecipitation analysis with nuclear extracts of E10.5 WT embryos revealed that Fam60a was present in immunoprecipitates prepared with antibodies to Sin3a or to Hdac1 but not in those prepared with antibodies to Hdac2 (*Figure 1B*), suggesting that the association between Hdac2 and Fam60a is relatively weak. Together, these data indicated that Fam60a is a component of the Sin3a-Hdac corepressor complex in developing mouse embryos and in undifferentiated P19 cells. This is consistent with recent findings that Fam60a is a core subunit of a variant Sin3a complex in ES cells (*Streubel et al., 2017*). Formation of the Sin3a-Hdac complex was not affected by the absence of Fam60a, however, given that Hdac1, Hdac2, and RbAp46/48 were co-immunoprecipitated with Sin3a from $Fam60a^{-/-}$ ES cells (*Figure 1—figure supplement 3*).

## *Fam60a* expression in mouse embryos and adult intestine

To shed light on the physiological function of Fam60a, we first examined the pattern of *Fam60a* expression during mouse embryogenesis. Expression of *Fam60a* was ubiquitous at E9.5, but it gradually became restricted to a subset of cells as development proceeded (*Figure 2—figure supplement 1*). At E12.5, *Fam60a* expression was thus apparent in the neural tube, neural crest cells, lung, pancreas, and intestine, but not in liver. Epithelial cells of the intestinal tract showed a high level of *Fam60a* expression at E15.5 (*Figure 2A*), and intervilli of the intestinal tract continued to express *Fam60a* at E17.5 (*Figure 2B and C*). In adult mice, Fam60a expression was maintained in crypts of the duodenum (*Figure 2D–F*). Given that intestinal stem and progenitor cells reside in crypts, we examined the fate of Fam60a$^+$ cells in crypts by administering tamoxifen to adult mice harboring a *Fam60a-CreERT2* transgene and *lacZ* reporter gene. Examination of the mice at 1, 3, and 5 days after tamoxifen injection revealed that LacZ$^+$ cells were present at the base of intestinal villi at 1 day and that they subsequently migrated toward the tip of the villi during the next 4 days (*Figure 2G–I*). These data thus suggested that *Fam60a* is expressed in a subset of cells including somatic stem cells in the intestine.

## Developmental defects in *Fam60a* mutant mice

We next generated mice lacking *Fam60a*. Two types of mutant allele were generated: $Fam60a^-$ and $Fam60a^{\beta geo}$ (*Figure 3—figure supplement 1A and B*). $Fam60a^{-/-}$ and $Fam60a^{\beta geo/\beta geo}$ mice showed indistinguishable phenotypes, suggesting that both alleles are functionally null, with subsequent analyses being performed with $Fam60a^{-/-}$ mice unless indicated otherwise. Both types of heterozygote also appeared indistinguishable from WT mice. We confirmed that *Fam60a* mRNA and Fam60a protein were absent in $Fam60a^{-/-}$ embryos (*Figure 3—figure supplement 1C and D*, *Figure 3—source data 1*). $Fam60a^{-/-}$ mice were born at a frequency much lower than that expected. They were detected at the expected frequency at E9.5 and E10.5, but their number started to decline thereafter and was greatly decreased at E18.5 (*Supplementary file 1*). Examination of $Fam60a^{-/-}$ embryos at E13.5 revealed that many visceral organs including the heart, lungs, liver, and gut were markedly smaller than those of WT embryos (*Figure 3A–C*). In particular, hypoplasia of the right ventricle of the heart was apparent, and a ventricular septum defect was also frequently observed, in $Fam60a^{-/-}$ embryos (*Figure 3D*). *Fam60a* was expressed in the developing heart, predominantly in the right ventricle and outflow tract, of WT embryos at E13.5 (*Figure 3G*). Many of the $Fam60a^{-/-}$

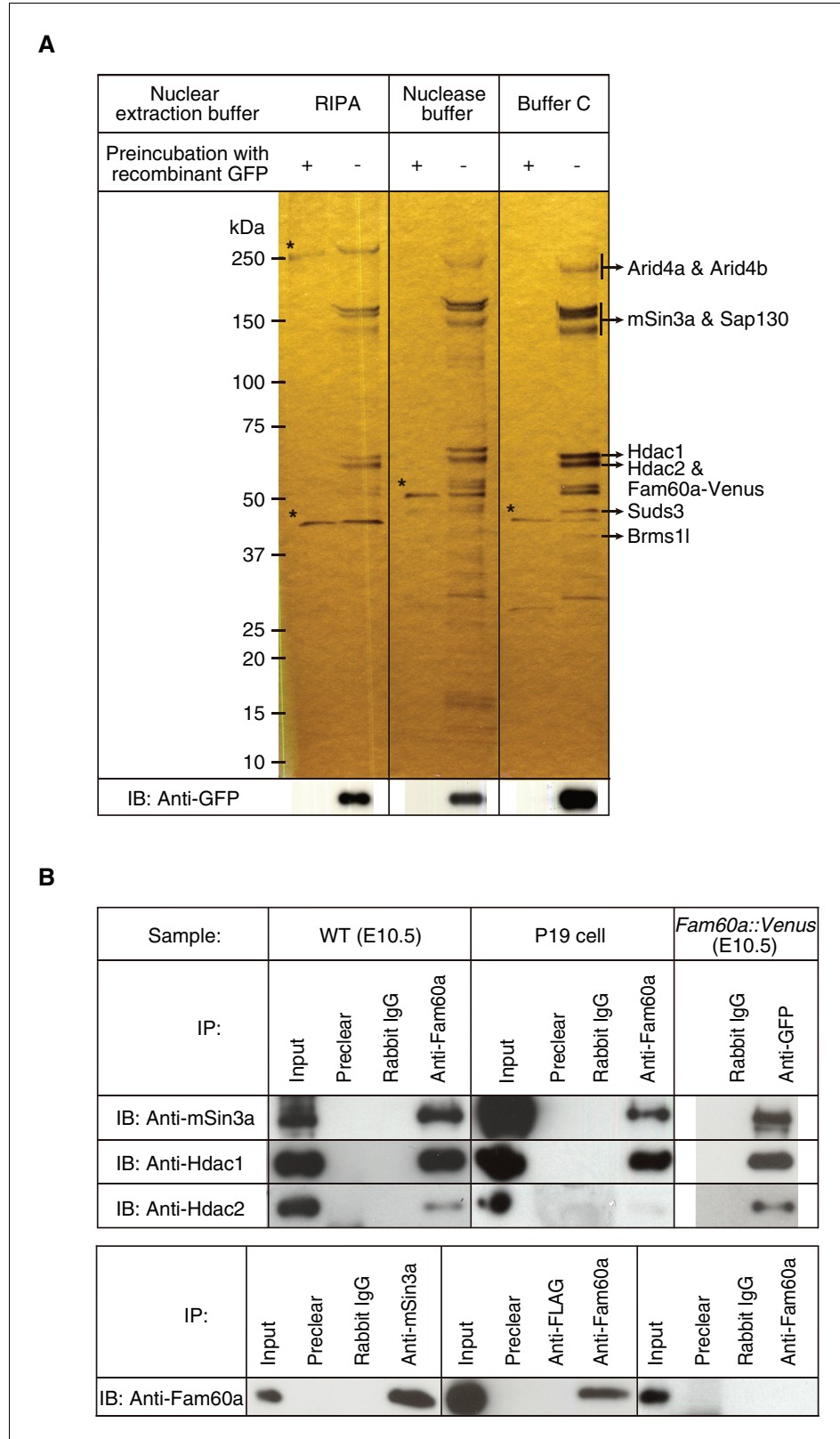

**Figure 1.** Identification of Fam60a-interacting proteins in mouse embryos. (**A**) Nuclear lysates prepared under three different conditions from E10.5 embryos harboring the *Fam60a::Venus* transgene were subjected to immunoprecipitation with bead-coupled antibodies to GFP that either had (control) or had not been previously exposed to recombinant GFP. Proteins that bound to the beads were then fractionated by SDS-polyacrylamide

*Figure 1 continued on next page*

*Figure 1 continued*

gel electrophoresis and revealed by silver staining. Proteins that bound nonspecifically to the beads are indicated by asterisks, and the identity of those that bound specifically was determined by mass spectrometry. The bead-bound proteins were also subjected to immunoblot (IB) analysis with antibodies to GFP for detection of Fam60a-Venus. (B) Nuclear extracts of E10.5 WT embryos, undifferentiated P19 cells, or E10.5 embryos harboring the *Fam60a::Venus* transgene were subjected to immunoprecipitation (IP) with bead-coupled anti-Fam60a, anti-GFP, or control rabbit immunoglobulin G (IgG), as indicated, and the resulting precipitates were subjected to immunoblot analysis with antibodies to Sin3a, to Hdac1, and to Hdac2 (upper panel). The nuclear extracts (Input) as well as the material that bound nonspecifically to beads before exposure to the antibodies used for immunoprecipitation (Preclear) were also subjected to immunoblot analysis. Alternatively, nuclear extracts of E10.5 WT embryos were subjected to immunoprecipitation with antibodies to Sin3a, to Hdac1, to Hdac2, or to the FLAG epitope (control), and the resulting precipitates were subjected to immunoblot analysis with antibodies to Fam60a (lower panel). See also *Figure 3—figure supplement 1 to 3*.

DOI: https://doi.org/10.7554/eLife.36435.003

The following figure supplements are available for figure 1:

**Figure supplement 1.** The *Fam60a::Venus* transgene encodes a functional Fam60a protein.
DOI: https://doi.org/10.7554/eLife.36435.004

**Figure supplement 2.** Immunofluorescence localization of Fam60a and Fam60a-Venus in P19 cells and mouse embryos.
DOI: https://doi.org/10.7554/eLife.36435.005

**Figure supplement 3.** Formation of the Sin3a-Hdac complex in the absence of Fam60a.
DOI: https://doi.org/10.7554/eLife.36435.006

embryos that survived to E18.5 manifested transposition of the great arteries, double-outlet right ventricle, and ventricular septum defects as well as spleen hypoplasia, incomplete lobulation of the lungs, and abnormal rotation of the gut (*Figure 3—figure supplements 2* and *3*). Although these abnormalities appeared reminiscent of laterality defects, left-right asymmetric expression of *Pitx2* was maintained at E8.0 (data not shown), suggesting that the abnormalities are not directly due to impaired left-right patterning.

The *Fam60a$^{-/-}$* embryos already showed morphological abnormalities including growth retardation as well as cardiac (shortening of the outflow tract) and neural tube defects at E9.5 (*Figure 3H and I*). Given that most of the mutant embryos manifested growth retardation, we examined the rate of cell proliferation in various tissues of embryos at E9.0 to E9.5 by labeling with bromodeoxyuridine (BrdU) and counting of BrdU-positive cells (*Figure 3—figure supplement 4*, *Figure 3—source data 2*). The extent of cell proliferation was significantly reduced in the septum transversum, secondary heart field (SHF), and proepicardium, whereas it was unaffected in the heart ventricle and slightly increased in the neural tube, of *Fam60a$^{-/-}$* embryos compared with control embryos. Given that the outflow tract is derived from SHF cells (*Buckingham et al., 2005*) and that *Fam60a* is expressed in SHF-derived regions of WT embryos (*Figure 3G*), the reduced proliferation rate of SHF cells may give rise to the shortening of the outflow tract and subsequent right ventricle hypoplasia apparent in the mutant embryos. These results thus suggested that *Fam60a* is required for cell proliferation and organogenesis in mouse embryos.

## Fam60a is recruited to promoter regions and regulates gene expression

Given that the Sin3a-Hdac complex is thought to repress gene expression by binding to promoter regions, we examined the global gene expression pattern in *Fam60a$^{-/-}$* embryos by RNA-sequencing (RNA-seq) analysis. Comparison of *Fam60a$^{-/-}$* and WT embryos at E9.5 revealed that the expression of 558 genes was up-regulated and that of 172 genes was down-regulated in the mutant embryos (*Figure 4A and B*, *Figure 4—source datas 1* and *2*). Gene ontology analysis revealed that the expression of genes related to the response to nutrients or to extracellular matrix organization was increased, whereas that of those related to lipid biosynthesis was decreased, in the mutant embryos (*Figure 4—figure supplement 1*). These data suggested that Fam60a regulates gene expression in both a negative and positive manner, but predominantly in a negative manner, in E9.5 embryos.

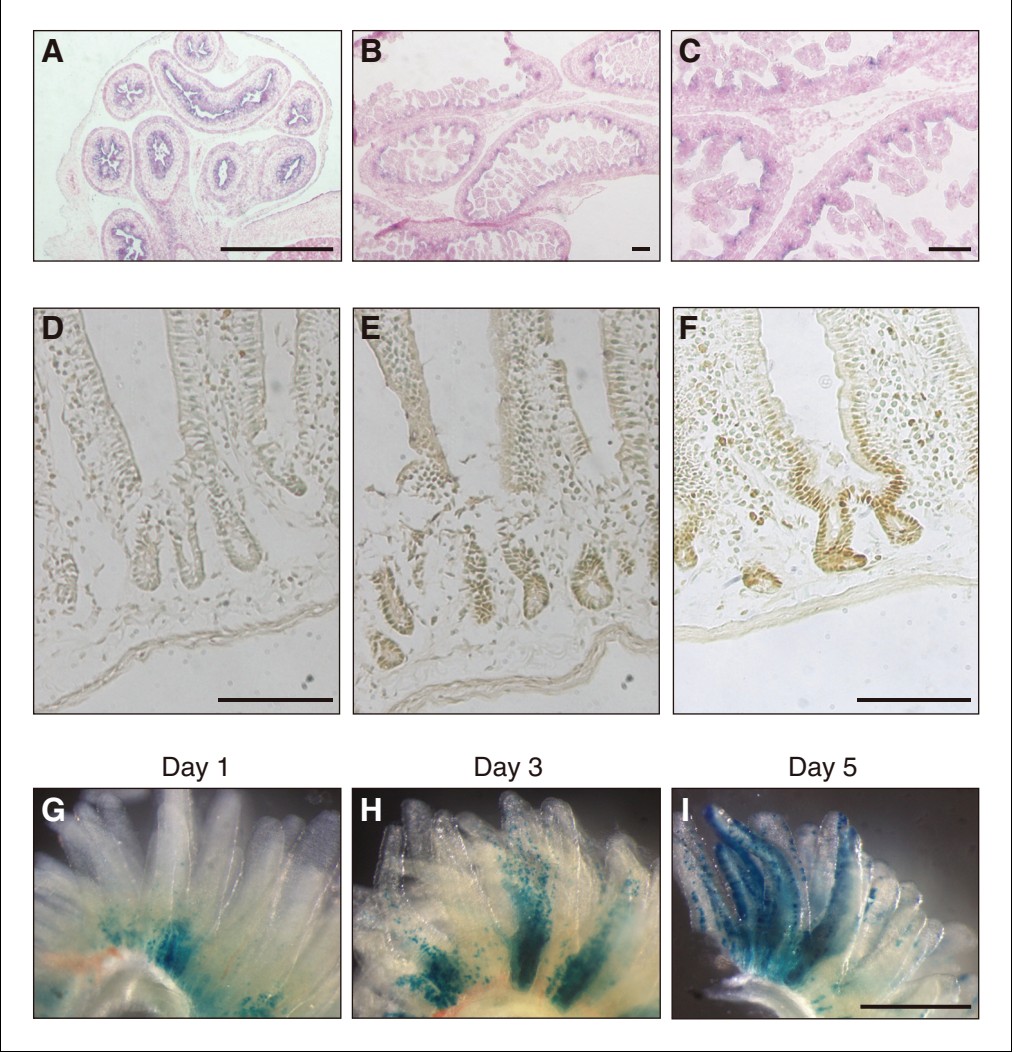

**Figure 2.** Expression of *Fam60a* in embryonic and adult mouse intestine. (**A–C**) In situ hybridization analysis of *Fam60a* expression in sections of mouse embryonic intestine at E15.5 (**A**) and E17.5 (**B** and **C**). *Fam60a* is expressed in epithelial cells of the gastrointestinal tract at E15.5 and in intervilli of the intestine at E17.5. A higher magnification view of the image in (**B**) is shown in (**C**). Scale bars, 100 µm. (**D–F**) Immunohistochemical analysis of Fam60a expression in the adult duodenum. Staining for a wild-type mouse was performed without primary antibodies as a control (**D**) or with antibodies to Fam60a (**E**), and that for a mouse harboring a *Fam60a::Venus* transgene was performed with antibodies to GFP (**F**). Scale bars, 100 µm. (**G–I**) Lineage trace analysis of LacZ[+] cells (stained with the LacZ substrate X-gal) in intestinal villi of the duodenum at 1, 3, or 5 days, respectively, after injection of tamoxifen (6 mg) in adult mice harboring a *Fam60a-CreERT2* transgene and *lacZ* reporter gene. Scale bar, 500 µm. See also *Figure 1—figure supplement 1*.

DOI: https://doi.org/10.7554/eLife.36435.007

The following figure supplement is available for figure 2:

**Figure supplement 1.** Expression of *Fam60a* in postimplantation mouse embryos.

DOI: https://doi.org/10.7554/eLife.36435.008

We also performed chromatin immunoprecipitation followed by deep sequencing (ChIP-seq) analysis with E9.5 *Fam60a::Venus* transgenic embryos and antibodies to GFP to identify Fam60a binding sites in the genome. The Fam60a-Venus fusion protein encoded by this transgene was able to rescue the defects of *Fam60a* mutant mice (*Figure 1—figure supplement 1B and C*), suggesting that it is fully functional. Approximately 17,000 and 14,000 peaks were detected in two independent experiments (ChIP-seq1 and ChIP-seq2, respectively), with ~80% of the peaks being localized at gene loci,

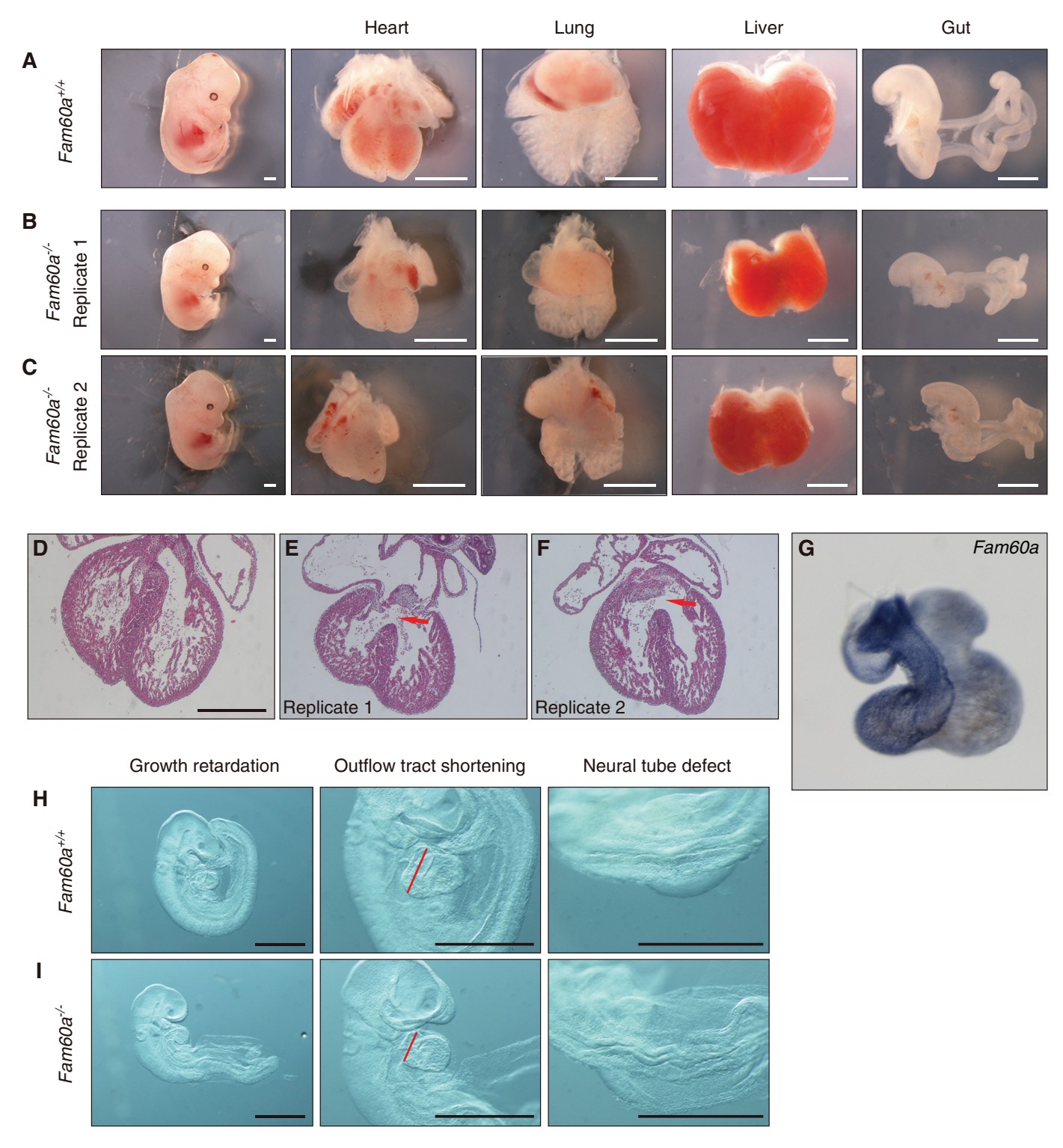

**Figure 3.** Growth retardation of visceral organs in *Fam60a*$^{−/−}$ mice. (A–C) Whole embryos and the indicated organs of WT (*Fam60a*$^{+/+}$, (A) and *Fam60a*$^{−/−}$ (B and C) mice at E13.5. Scale bars, 1 mm. (D–F) Sections of the heart of WT (D) or *Fam60a*$^{−/−}$ (E and F) embryos at E13.5 stained with hematoxylin-eosin. The mutant embryos manifest ventricular septum defects (red arrows). Scale bar, 500 μm. (G) Expression of *Fam60a* in E13.5 heart was examined by whole-mount in situ hybridization. (H and I) *Fam60a*$^{−/−}$ and WT embryos, respectively, at E9.5. The mutant embryos show overall growth retardation as well as shortening of the outflow tract (red bars) and a severe neural tube defect. Scale bars, 1 mm. See also *Figure 3—figure supplement 1 to 4* and *Supplementary file 1*.

*Figure 3 continued on next page*

*Figure 3 continued*

DOI: https://doi.org/10.7554/eLife.36435.009

The following source data and figure supplements are available for figure 3:

**Source data 1.** Numerical data of *Figure 3—figure supplement 1C*.

DOI: https://doi.org/10.7554/eLife.36435.014

**Source data 2.** Numerical data of *Figure 3—figure supplement 4I*.

DOI: https://doi.org/10.7554/eLife.36435.015

**Figure supplement 1.** Generation of *Fam60a* mutant mice.

DOI: https://doi.org/10.7554/eLife.36435.010

**Figure supplement 2.** Impaired organogenesis in *Fam60a$^{-/-}$* embryos at E18.5.

DOI: https://doi.org/10.7554/eLife.36435.011

**Figure supplement 3.** Gut looping defect in *Fam60a$^{-/-}$* mice.

DOI: https://doi.org/10.7554/eLife.36435.012

**Figure supplement 4.** BrdU immunohistochemistry for determination of the proliferation index in *Fam60a$^{-/-}$* and WT embryos.

DOI: https://doi.org/10.7554/eLife.36435.013

in particular in the vicinity of transcription start sites (TSSs) (*Figure 5A and B*; *Figure 5—figure supplement 1A and B*, *Figure 5—source data 1*). This distribution pattern was highly similar to that previously determined for Sin3a (*Bowman et al., 2014*). Co-immunoprecipitation analysis of E10.5 transgenic embryos revealed that Fam60a-Venus interacts with Ing2 (*Figure 5—figure supplement 1C*), a protein that binds to Lys$^4$-trimethylated histone H3 (H3K4me3), suggesting that Fam60a is recruited predominantly to the promoters of transcribed genes. Examination of the TSS region (between –3 kb and +3 kb relative to the TSS) of all genes resulted in the identification of 7989 genes that reproducibly showed at least one Fam60a binding site in this region (*Figure 5C*), suggesting that these genes may be directly regulated by Fam60a.

Among the 558 up-regulated and 172 down-regulated genes identified in *Fam60a$^{-/-}$* embryos, 245 and 45 genes, respectively, had at least one Fam60a binding peak in the TSS region (*Figure 4A*). Given that 74% (127/172) of the down-regulated genes lacked a Fam60a binding site in this region, the change in expression of most of the down-regulated genes was likely due to a secondary effect of Fam60a loss. We selected for further analysis 18 genes from the 290 (245 + 45) identified genes on the basis of their large fold change in expression in the mutant embryos as revealed by RNA-seq (*Figure 4B*). Reverse transcription and quantitative polymerase chain reaction (RT-qPCR) analysis confirmed significant differences in expression level for at least six of these putative Fam60a target genes between WT and *Fam60a$^{-/-}$* embryos at E9.5, with the expression of *Leng9*, *Adhfe1*, *Mxd3*, *Dchs1*, and *Nagk* being up-regulated and that of *Gt(ROSA)26Sor* being down-regulated in the mutant (*Figure 4C*, *Figure 4—source data 3*). The expression of some of these up-regulated genes (such as *Leng9*, *Dchs1*, and *Nagk*) was also increased in *Fam60a$^{-/-}$* ES cells compared with control ES cells (*Figure 4—figure supplement 2A and B*, *Figure 4—source data 4*). ChIP-qPCR analysis for three of the up-regulated genes (*Adhfe1*, *Nagk*, *Dchs1*) also revealed the association of their promoter regions with Fam60a-Venus and Sin3a in E9.5 transgenic and WT embryos, respectively (*Figure 5D*, *Figure 5—source data 2*). The Fam60a binding peaks identified by ChIP-seq analysis in the TSS regions of *Adhfe1*, *Nagk*, and *Dchs1* are shown in *Figure 5E*. Although the Sin3a-Hdac complex possesses histone-deacetylating activity, the level of Lys$^9$-acetylated histone H3 (AcH3K9) at the promoter regions of Fam60a target genes (*Adhfe1*, *Nagk*, *Dchs1*) did not differ between WT and *Fam60a$^{-/-}$* embryos (*Figure 5—figure supplement 2*, *Figure 5—source data 3*). However, similar analysis with *Fam60a$^{-/-}$* ES cells revealed that the level of AcH3K9 at the promoter regions of three such genes (*Leng9*, *Dchs1*, *Nagk*) was increased (*Figure 4—figure supplement 2C*, *Figure 4—source data 5*).

## Association of Fam60a with DNA methylation and Tet

We examined the molecular phylogeny of *Fam60a* with a sequence data set containing invertebrate homologs as well as a paralog, designated *Fam60b* (*Figure 6A*). The phylogenetic tree revealed the gene duplication event that gave rise to *Fam60a* and *Fam60b* in the early vertebrate lineage before the radiation of jawed vertebrates, likely during the well-studied genome expansion (2R-WGD, two-

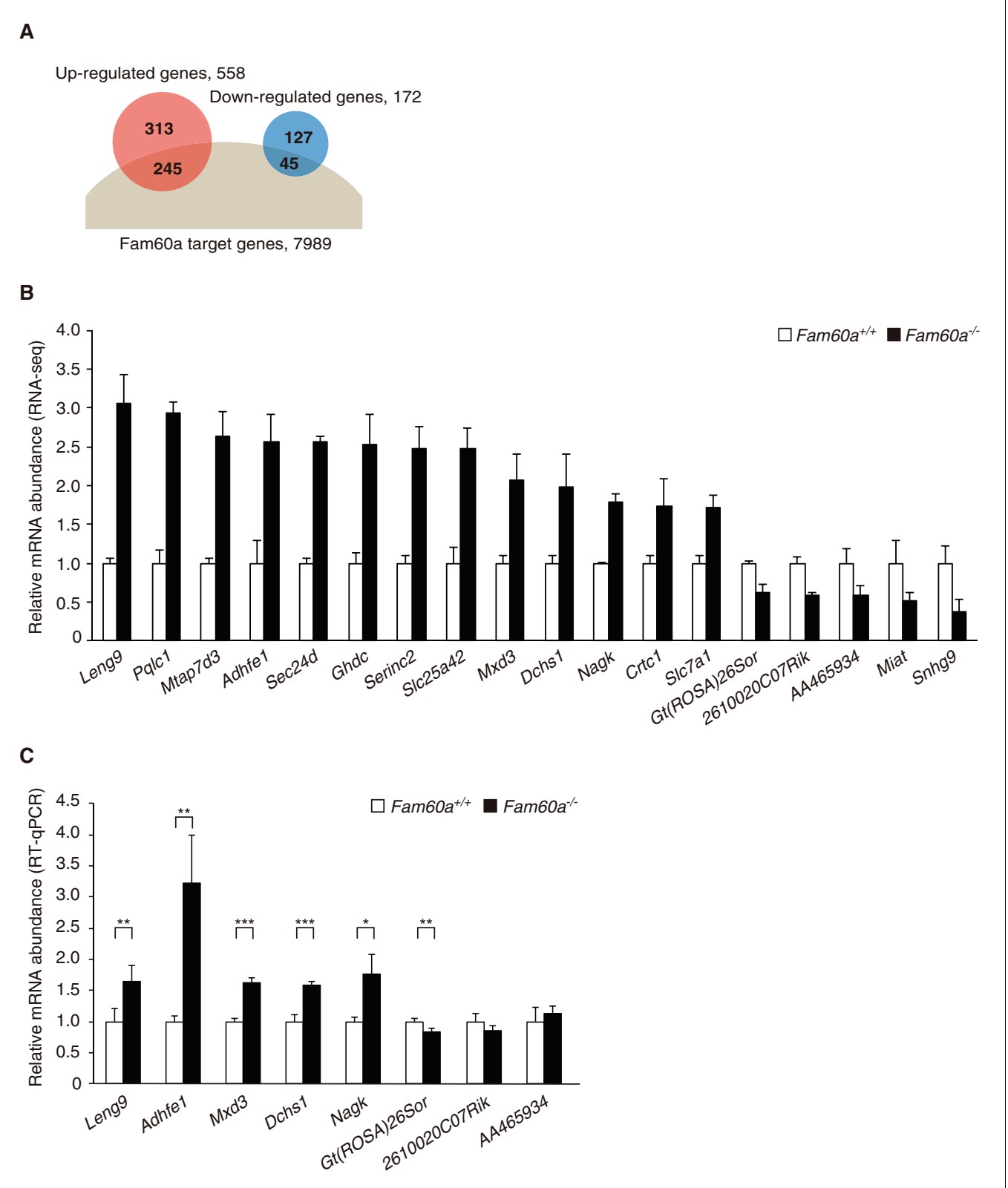

**Figure 4.** Altered gene expression profile in *Fam60a*$^{-/-}$ embryos. (**A**) Venn diagram showing the overlap between Fam60a target genes identified by ChIP-seq analysis and genes whose expression was up- or down-regulated in *Fam60a*$^{-/-}$ embryos at E9.5 as revealed by RNA-seq analysis. (**B**) Fold change in RNA-seq values for genes that were differentially expressed in E9.5 *Fam60a*$^{-/-}$ relative to WT embryos and which were also found to bind Fam60a-Venus in the TSS region by the ChIP-seq analysis. Data are means ± s.d. for three embryos. (**C**) Validation of RNA-seq data by RT-qPCR analysis

*Figure 4 continued on next page*

*Figure 4 continued*

for E9.5 WT and *Fam60a*$^{-/-}$ embryos. Data are means ± s.d. for five independent experiments. *p<0.05, **p<0.01, ***p<0.001 (Student's unpaired *t* test). See also *Figure 4—figure supplements 1* and *2*.

DOI: https://doi.org/10.7554/eLife.36435.016

The following source data and figure supplements are available for figure 4:

**Source data 1.** List of target genes of *Fam60a*.
DOI: https://doi.org/10.7554/eLife.36435.019
**Source data 2.** Numerical data of *Figure 4B*.
DOI: https://doi.org/10.7554/eLife.36435.020
**Source data 3.** Numerical data of *Figure 4C*.
DOI: https://doi.org/10.7554/eLife.36435.021
**Source data 4.** Numerical data of *Figure 4—figure supplement 2B*.
DOI: https://doi.org/10.7554/eLife.36435.022
**Source data 5.** Numerical data of *Figure 4—figure supplement 2C*.
DOI: https://doi.org/10.7554/eLife.36435.023
**Source data 6.** Numerical data of *Figure 4—figure supplement 2D*.
DOI: https://doi.org/10.7554/eLife.36435.024
**Figure supplement 1.** Gene ontology analysis of differentially expressed genes in *Fam60a*$^{-/-}$ embryos at E9.5.
DOI: https://doi.org/10.7554/eLife.36435.017
**Figure supplement 2.** Expression profile as well as AcH3K9 deposition and Tet1 recruitment at promoter regions of Fam60a target genes in mouse ES cells.
DOI: https://doi.org/10.7554/eLife.36435.018

round whole genome duplication) that occurred in this period. It also highlighted the origin of the preduplication ortholog *Fam60* in the early metazoan era. Analysis of the families of genes encoding Sin3, Tet, and Dnmt proteins as well as the presence or absence of DNA methylation in individual species suggested an association of Fam60a with DNA methylation, Tet, and Sin3 (*Figure 6B*). Fam60a proteins of ~220 amino acid residues were thus found in all vertebrates examined, and Fam60a orthologs were also detected in insects but not in nematodes or yeasts (*Figure 6—figure supplement 1*) (*Smith et al., 2012*). DNA methylation and Tet proteins are also conserved from humans to insects but not in nematodes or yeasts, whereas Sin3 is more widely conserved from yeasts to humans.

The association of Fam60a with DNA methylation and Tet, together with the fact that the Sin3a-Hdac complex interacts with methylation-regulating proteins such as methylated CpG binding protein2 (MeCP2), Dnmt1, and Tet1 (*Nan et al., 1998*; *Williams et al., 2011*), suggested that Fam60a might regulate Tet-mediated DNA demethylation. We tested this possibility in NIH3T3 cells transfected with a doxycycline-inducible expression vector for FLAG epitope–tagged Tet1 and with either an expression vector for both Fam60a and Venus or the corresponding empty vector. Exposure of the transfected cells to doxycycline thus induced the expression of Tet1 in the absence or presence of that of Fam60a (*Figure 6—figure supplement 2*). In the absence of Fam60a, 83% of FLAG-Tet1$^{+}$ cells were positive for 5hmC (that is, only 17% of FLAG-Tet1$^{+}$ cells remained negative for 5hmC) at 24 hr after the administration of doxycycline, suggestive of the efficient conversion of 5mC to 5hmC by FLAG-Tet1. In the presence of Fam60a, however, 55% of FLAG-Tet1$^{+}$ cells remained negative for 5hmC (*Figure 6C–F*, *Figure 6—source data 1* and *2*), suggesting that Fam60a might inhibit Tet1 activity. Recruitment of Tet1 to the promoter regions of Fam60a target genes (*Leng9*, *Dchs1*, *Nagk*) was not affected in *Fam60a*$^{-/-}$ ES cells (*Figure 4—figure supplement 2D*, *Figure 4—source data 6*), suggesting that Fam60a negatively regulates Tet1 activity without affecting its recruitment to promoter regions.

## Aberrant promoter hypomethylation in *Fam60a*$^{-/-}$ mouse embryos

Given that our results suggested that Fam60a inhibits Tet1 activity in cultured cells, we next determined whether DNA methylation is affected in *Fam60a*$^{-/-}$ mouse embryos. Bisulfite sequencing of the promoter regions of *Nagk* and *Leng9* revealed little or no DNA methylation in WT or *Fam60a*$^{-/-}$ embryos at E9.5 (*Figure 7—figure supplement 1*), even though our ChIP analyses showed that

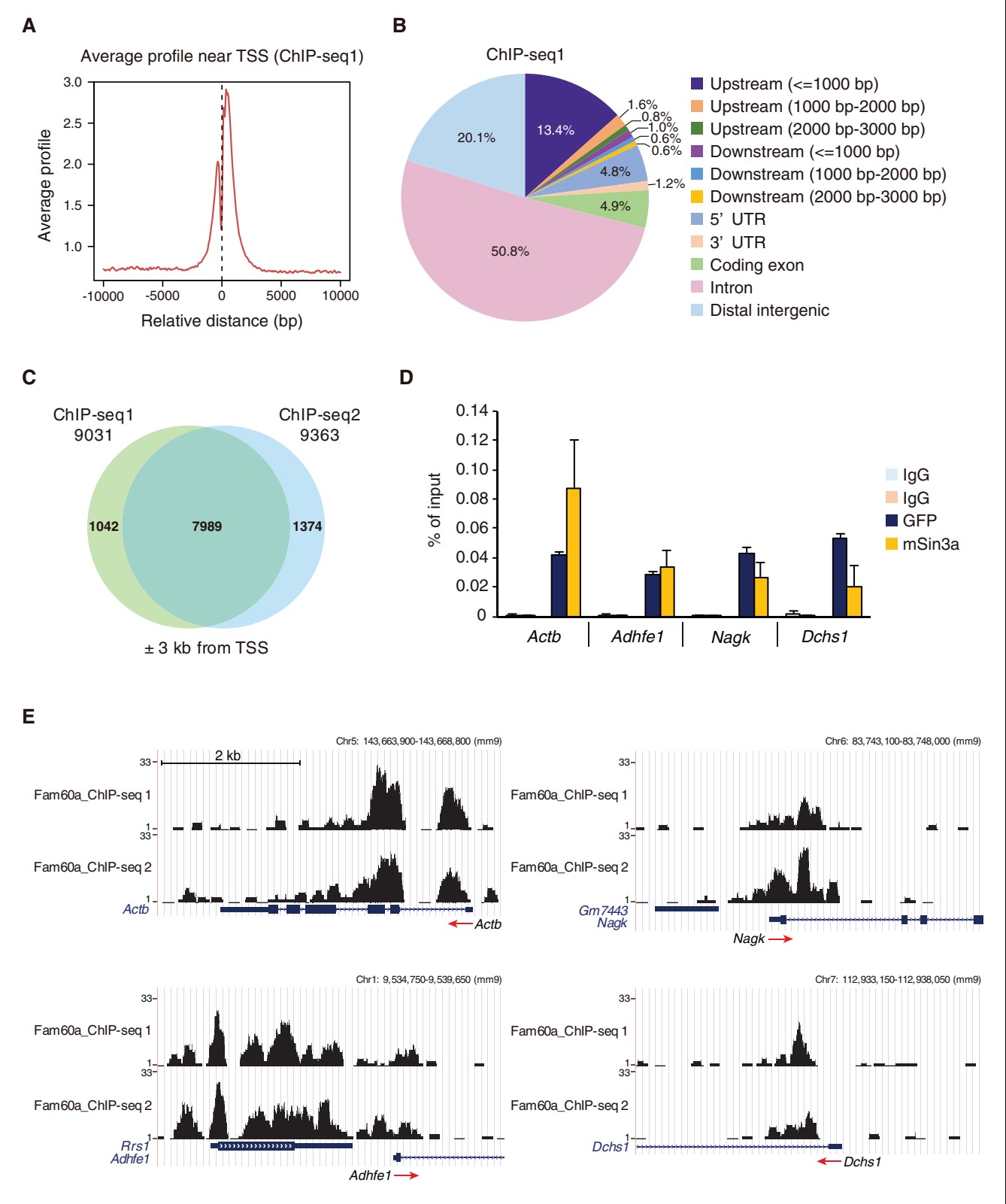

**Figure 5.** Genome-wide localization of Fam60a to gene promoters. (**A**) Average binding profile for Fam60a-Venus at the TSS region of all genes with binding peaks identified by ChIP-seq analysis of E9.5 transgenic embryos with antibodies to GFP. Distance is expressed relative to the TSS. (**B**) Peak distribution for ChIP-seq analysis as in (**A**). About 80% of peaks were localized to gene loci. UTR, untranslated region. (**C**) Venn diagram showing the overlap in Fam60a target genes (those with binding peaks within ±3 kb of the TSS) for two independent ChIP-seq analyses (ChIP-seq1 and ChIP-seq2). *Figure 5 continued on next page*

Figure 5 continued

(**D**) ChIP-qPCR analysis of the binding of Fam60a-Venus and Sin3a to the TSS regions of the indicated genes in E9.5 transgenic and WT embryos, respectively. The pale blue and orange bars represent IgG controls for antibodies to GFP and to Sin3a, respectively. Data are expressed as percentage of input and are means ± s.d. for three independent experiments. *Actb* was examined as a positive control. (**E**) Examples of Fam60a-Venus ChIP-seq results for E9.5 *Fam60a-Venus* embryos. ChIP-seq1 and ChIP-seq2 were both performed with antibodies to GFP. Peaks around the TSS are shown for four Fam60a target genes, with red arrows indicating the direction of transcription. See also *Figure 5—figure supplements 1* and *2*.
DOI: https://doi.org/10.7554/eLife.36435.025

The following source data and figure supplements are available for figure 5:

**Source data 1.** List of target genomic regions identified by ChIP-seq analysis.
DOI: https://doi.org/10.7554/eLife.36435.028
**Source data 2.** Numerical data of *Figure 5D*.
DOI: https://doi.org/10.7554/eLife.36435.029
**Source data 3.** Numerical data of *Figure 5—figure supplement 2*.
DOI: https://doi.org/10.7554/eLife.36435.030
**Figure supplement 1.** Genome-wide localization of Fam60a to gene promoters.
DOI: https://doi.org/10.7554/eLife.36435.026
**Figure supplement 2.** ChIP-qPCR analysis of AcH3K9 at Fam60a target gene promoters.
DOI: https://doi.org/10.7554/eLife.36435.027

Fam60a-Venus was recruited to these promoter regions in transgenic embryos. In contrast, the promoter region of *Adhfe1* was found to be hypomethylated in *Fam60a*$^{-/-}$ embryos, with a methylation level of 4 to 10% compared with a value of ~20% in WT embryos at E9.5 (*Figure 7*, *Figure 7—source data 1*). This hypomethylation might have been due to reduced de novo DNA methylation or increased demethylation mediated by Tet. To distinguish between these possibilities, we examined methylation of the *Adhfe1* promoter at earlier developmental stages, given that de novo DNA methylation occurs predominantly before implantation. No significant difference in methylation was observed between WT and *Fam60a*$^{-/-}$ embryos at E7.5, after which the methylation level of this promoter gradually decreased in the mutant embryos (*Figure 7*, *Figure 7—source data 1*). These results suggested that impaired maintenance of methylation or increased demethylation is responsible for the hypomethylation of the *Adhfe1* promoter in *Fam60a*$^{-/-}$ embryos, consistent with our observation that Fam60a inhibited Tet1 activity in cultured cells. Providing further support for this notion, hydroxymethyl DNA immunoprecipitation (hMeDIP) analysis revealed 5hmC deposition at almost all Fam60a target gene promoters examined in WT embryos (*Figure 7—figure supplement 2*, *Figure 7—source data 2*). Hypomethylation was not detected at the imprinting control regions of *Kcnq1ot1* or *Peg3* in *Fam60a*$^{-/-}$ embryos (*Figure 7—figure supplement 3*). Together, these findings suggested that Fam60a regulates Tet-mediated demethylation at a subset of gene promoters.

## Differentially methylated regions in the genome of *Fam60a*$^{-/-}$ embryos

To verify the role of Fam60a in regulation of DNA methylation, we examined the methylation status of promoters, CpG islands, and CpG shores in the genome of *Fam60a*$^{-/-}$ and WT embryos at E9.5. These target regions were captured, subjected to bisulfite conversion, and sequenced with a next-generation sequencer. The overall methylation level of CpG sites in the captured DNA was around 45% and showed a similar distribution pattern in both *Fam60a*$^{-/-}$ and WT embryos (*Supplementary file 2*, *Figure 8—figure supplement 1*).

Given that genome-wide DNA methylation level did not appear to be affected by the absence of Fam60a, we first examined DNA methylation levels over Fam60a-bound promoters (~8000 promoters) in the wild-type and *Fam60a*$^{-/-}$ embryos. Hypomethylation was commonly observed at the Fam60a-binding regions, but there was no obvious difference in the profile between the wild-type and *Fam60a*$^{-/-}$ embryos (*Figure 8—figure supplement 2*). We next examined if the DNA methylation level was affected in a subset of gene promoters, by focusing on differentially methylated regions (DMRs). 7245 DMRs were detected with average changes of DNA methylation 11.87 and 10.99% for hyper- and hypomethylated DMRs, respectively (*Figure 8—figure supplement 3*). Among the 7245 DMRs detected, 3049 and 4196 regions were hyper- and hypomethylated, respectively, in *Fam60a*$^{-/-}$ embryos, with 388 hypermethylated DMRs (12.7%) and 1257 hypomethylated

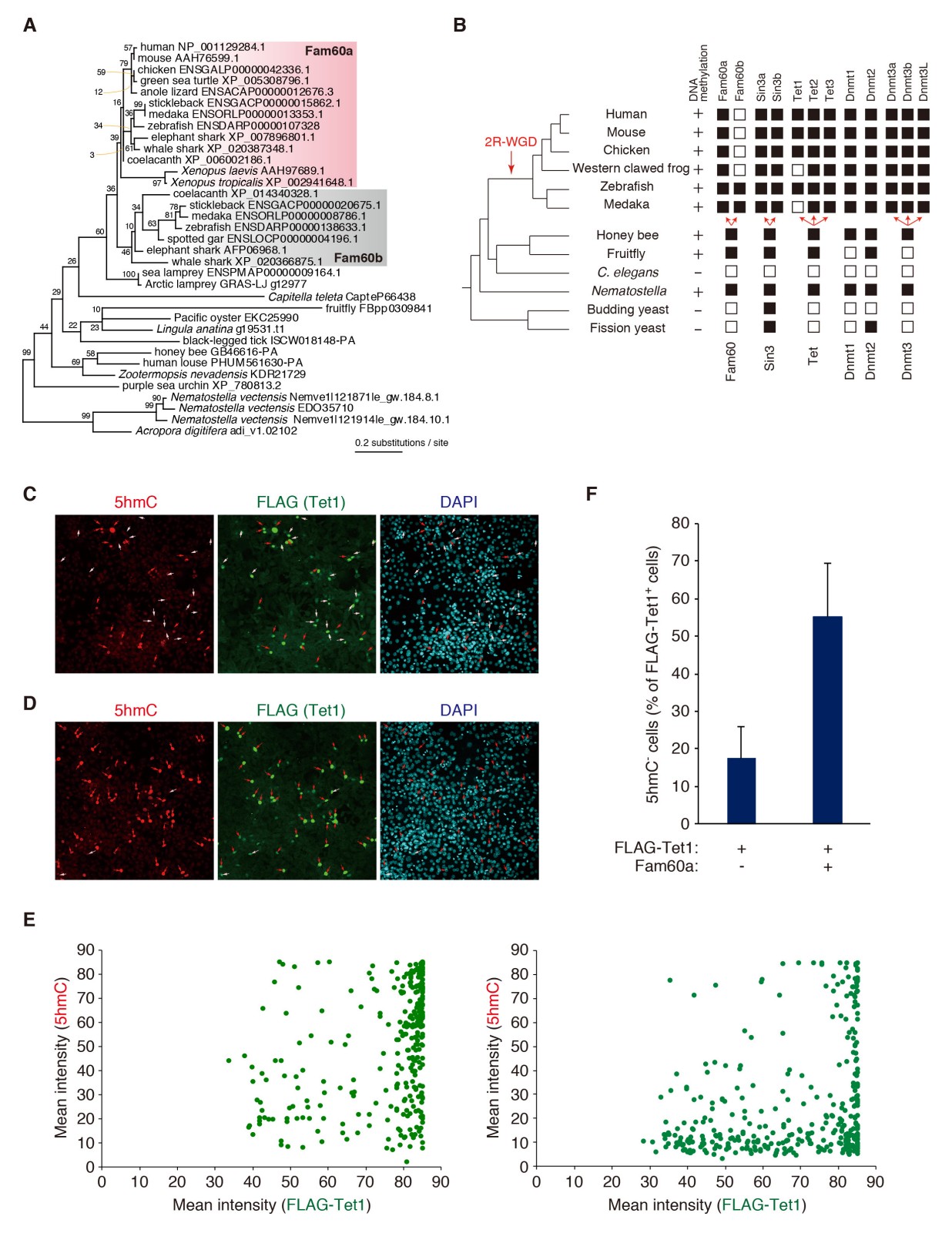

**Figure 6.** Phylogenetic and functional relation between Fam60a and Tet1. (**A**) Molecular phylogeny of Fam60a and related proteins. The tree was inferred with the maximum-likelihood method and 99 amino acid residues. Bootstrap values are indicated at individual nodes. (**B**) Gene repertories for Fam60, Sin3, Tet, and Dnmt families. Black boxes indicate the presence of at least one phylogenetically validated ortholog, whereas white boxes indicate the absence of orthologs. 2R-WGD, two rounds of whole-genome duplication. The presence or absence of DNA methylation in individual

*Figure 6 continued on next page*

*Figure 6 continued*

species based on current knowledge (*Suzuki and Bird, 2008*; *Zemach and Zilberman, 2010*) is also shown. (**C** and **D**) Fam60a inhibits Tet1 activity in NIH3T3 cells. Immunofluorescence staining of 5hmC and FLAG-Tet1 was performed for NIH3T3 cells expressing FLAG-Tet1 either together with Fam60a (**C**) or alone (**D**). The cells were analyzed 24 hr after the induction of FLAG-Tet1 expression by doxycycline administration. Nuclei were stained with 4',6-diamidino-2-phenylindole (DAPI). Red arrows indicate cells positive for both 5hmC and FLAG immunoreactivity. White arrows indicate cells positive for FLAG but negative for 5hmC. (**E**) Plots of mean fluorescence intensity for 5hmC versus FLAG-Tet1 in cells expressing FLAG-Tet1 without (left) or with (right) Fam60a as in (**C**) and (**D**). (**F**) Proportion of FLAG-Tet1[+] cells that were negative for 5hmC in experiments similar to that in (**C**) and (**D**). Data are means ± s.d. for three independent experiments. *p<0.05 (Student's unpaired *t* test). See also *Figure 6—figure supplements 1* and *2*.
DOI: https://doi.org/10.7554/eLife.36435.031

The following source data and figure supplements are available for figure 6:

**Source data 1.** Numerical data of *Figure 6E*.
DOI: https://doi.org/10.7554/eLife.36435.034
**Source data 2.** Numerical data of *Figure 6F*.
DOI: https://doi.org/10.7554/eLife.36435.035
**Figure supplement 1.** Alignment of the predicted amino acid sequences of Fam60a proteins by Clustal OMEGA.
DOI: https://doi.org/10.7554/eLife.36435.032
**Figure supplement 2.** Experimental strategy for expression of Fam60a and Venus and inducible expression of FLAG-Tet1 in NIH3T3 cells.
DOI: https://doi.org/10.7554/eLife.36435.033

DMRs (30.0%) being found to overlap with Fam60a binding regions (*Table 1*). Among the top 500 hyper- and hypomethylated DMRs showing the largest differences in methylation level between mutant and WT embryos, 83 of the hypermethylated DMRs (16.6%) and 254 of the hypomethylated DMRs (50.8%) contained Fam60a binding sites (*Table 1*), suggestive of a preferential association of Fam60a binding sites with hypomethylated DMRs. The promoter of *Adhfe1*, which was found to be hypomethylated in *Fam60a*[−/−] embryos (*Figure 7*), was included in the top 500 hypomethylated DMRs (*Figure 8—source data 1*).

We next examined the positions of the top 500 hypermethylated and top 500 hypomethylated DMRs in the genome. The distributions of these two types of region differed, with hypermethylated DMRs being preferentially located in exonic regions of genes at 5 to 50 kb downstream of the TSS (*Figure 8A and C*), whereas most hypomethylated DMRs were located in intronic regions at 0 to 5 kb downstream of the TSS (*Figure 8B and D*). The distribution pattern of hypomethylated DMRs (*Figure 8B*) was similar to that of Fam60a binding sites (*Figure 5B* and *Figure 5—figure supplement 1B*). These data thus suggested that Fam60a is associated with DNA methylation status in mouse embryos.

## Discussion

*Fam60a* is expressed ubiquitously during mouse embryonic development until at least E9.5, after which its expression gradually becomes restricted to a subset of cells, including those engaged in proliferation. In the adult mouse, *Fam60a* is expressed in stem cells located in intestinal crypts, suggesting that its expression may be associated with differentiation potential. Consistent with this notion, *Fam60a* knockout mice manifest growth retardation in visceral organs. Gene ontology analysis revealed that genes whose expression is dysregulated in *Fam60a*[−/−] embryos include those related to the response to nutrients, extracellular matrix organization, and lipid biosynthesis, suggesting that disruption of these processes may contribute to the retardation of organ growth apparent in the mutant embryos.

A search for Fam60a-interacting proteins identified the Sin3a-Hdac transcriptional corepressor complex. The stoichiometry of Fam60a and components of this complex recovered in immunoprecipitates (*Figure 1*) suggested that most Fam60a in a given cell is associated with the complex. Fam60a may therefore function in association with the Sin3a-Hdac complex. Whereas Sin3a knockout mice die during embryogenesis around the time of implantation (*McDonel et al., 2012*), *Fam60a*[−/−] embryos develop until later stages. It is thus possible that Sin3a has functions independent of Fam60a, including functions in multiple protein complexes, or that the earlier defects of Sin3a knockout mice are due to the lack of this protein in oocytes. Fam60a was recently shown to be a core subunit of a variant Sin3a complex in ES cells that includes Tet1 and Ogt (*Streubel et al., 2017*).

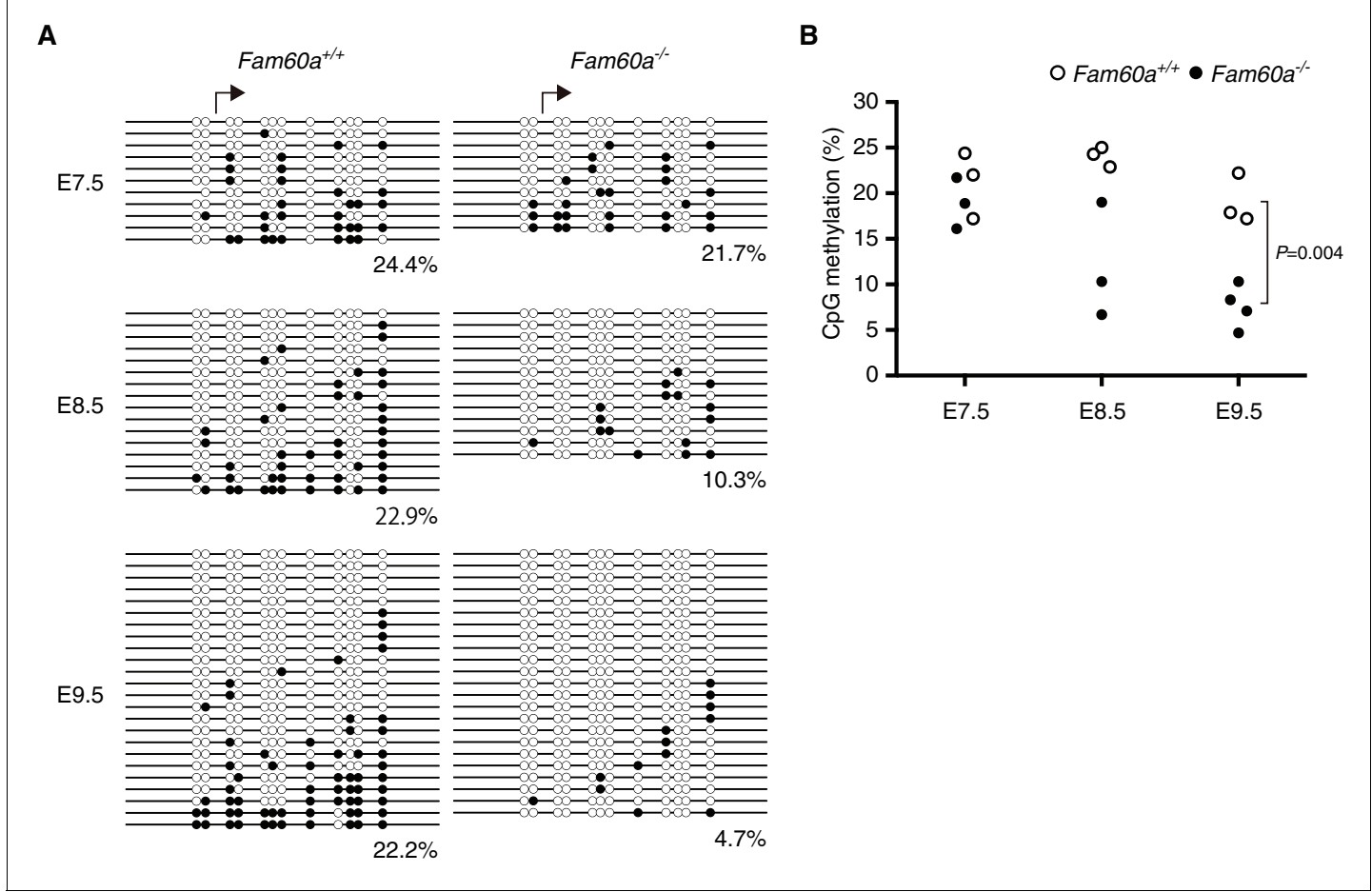

**Figure 7.** Methylation status of the *Adhfe1* promoter during development of WT and *Fam60a⁻/⁻* mouse embryos. (**A**) Methylation pattern at the *Adhfe1* promoter in representative WT and *Fam60a⁻/⁻* embryos at the indicated developmental stages as revealed by bisulfite sequencing. Closed and open circles indicate methylated and nonmethylated CpG sites, respectively. Arrows indicate the TSS of *Adhfe1*. (**B**) Methylation frequency at the *Adhfe1* promoter determined as in (**A**) for three or four individual embryos of each genotype at each developmental stage. The p value was determined with Student's unpaired *t* test. See also *Figure 7—figure supplement 1 to 3*.

DOI: https://doi.org/10.7554/eLife.36435.036

The following source data and figure supplements are available for figure 7:

**Source data 1.** Numerical data of *Figure 7B*.
DOI: https://doi.org/10.7554/eLife.36435.040

**Source data 2.** Numerical data of *Figure 7—figure supplement 2*.
DOI: https://doi.org/10.7554/eLife.36435.041

**Figure supplement 1.** Bisulfite sequencing of Fam60a target gene promoters.
DOI: https://doi.org/10.7554/eLife.36435.037

**Figure supplement 2.** Deposition of 5hmC at Fam60a target gene promoters as revealed by hMeDIP analysis in WT embryos at E9.5.
DOI: https://doi.org/10.7554/eLife.36435.038

**Figure supplement 3.** Methylation status of imprinting control regions as determined by bisulfite sequencing.
DOI: https://doi.org/10.7554/eLife.36435.039

In general, the Sin3a-Hdac complex is thought to repress gene expression via histone deacetylation. However, this complex can also facilitate transcriptional activation in a manner dependent on cellular context (*Suganuma and Workman, 2013*; *Icardi et al., 2012*). Indeed, we found that the expression of many genes was either up-regulated or down-regulated in *Fam60a⁻/⁻* embryos. Fam60a may therefore contribute not only to the transcriptional corepressor activity of the Sin3a-Hdac complex but also to its promotion of transcriptional activation. Fam60a likely does not serve as

**Table 1.** The number of hyper- and hypomethylated DMRs overlapping with ChIP-seq peaks.

Relation between DMRs and Fam60a binding site for E9.5 embryos. Methyl-seq data were obtained for three *Fam60a^-/-^* and three WT embryos, and ChIP-seq data were obtained for ChIP-seq1 and ChIP-seq2 experiments. The number of ChIP-seq peaks that overlap with all or the top 500 hyper- and hypomethylated DMRs are shown.

| Data set | Total DMRs | Direction | DMRs | Overlap with ChIP-seq peaks | |
|---|---|---|---|---|---|
| | | | | vs. all DMRs (%) | vs. top 500 (%) |
| 3 embryos (triplicates) (mean Diff >= 0.05) | 7245 | Hyper | 3049 | 388 (12.7) | 83 (16.6) |
| | | Hypo | 4196 | 1257 (30.0) | 254 (50.8) |

DOI: https://doi.org/10.7554/eLife.36435.042

a simple regulator of Hdac activity, given that the level of histone acetylation at Fam60a target gene promoters did not differ between WT and *Fam60a^-/-^* embryos.

Phylogenetic analysis revealed a wide taxonomic distribution of the ancestral *Fam60* gene in eumetazoans and a duplication of this gene during early vertebrate evolution that gave rise to *Fam60a* and *Fam60b* paralogs. The absence of *Fam60* and *Tet* genes as well as of DNA methylation in both *Caenorhabditis elegans* and yeasts suggests that Fam60a may contribute to Sin3a function related to DNA methylation and Tet. Consistent with this possibility, Sin3a is known to interact with MeCP2, Dnmt1, and Tet1 (*Nan et al., 1998*; *Williams et al., 2011*). Of note, Tet proteins play a role in demethylation of evolutionarily conserved gene enhancers during the phylotypic period of early development (*Bogdanović et al., 2016*). *Adhfe1*, whose promoter was found to be hypomethylated in *Fam60a^-/-^* embryos, appears to be a typical gene regulated by Fam60a and Tet activity. Expression of *Adhfe1* is thus normally repressed because of the methylation of its promoter that results from Fam60a-mediated inhibition of Tet activity, but it is up-regulated in *Fam60a^-/-^* embryos because of the promoter hypomethylation that results from the absence of Fam60a. Other genes whose expression was up-regulated in *Fam60a^-/-^* embryos (such as *Leng9* and *Nagk*) showed almost no DNA methylation in their promoter regions in either WT or mutant embryos, even though Fam60a-Venus was efficiently recruited to these promoters in transgenic embryos. In addition to functioning as DNA demethylases, Tet proteins associate with Sin3a-Hdac and act as transcriptional repressors in a manner independent of their demethylating activity (*Williams et al., 2011*; *Zhang et al., 2015*). Up-regulation of genes such as *Leng9* and *Nagk* in *Fam60a^-/-^* embryos may thus be due to the lack of the latter function of Tet proteins.

A genome-wide search for Fam60a binding sites revealed that Fam60a is recruited to gene promoter regions that overlap with CpG islands. In general, such CpG island promoters of transcriptionally active genes are enriched in H3K4me3. Consistent with the genomic localization of Fam60a, we found that Fam60a interacts with Ing2, which is known to bind to H3K4me3 (*Goeman et al., 2008*). Tet proteins interact with Sin3a and are thought to localize to CpG island promoters in order to maintain the CpG islands unmethylated, with such promoters often being marked with H3K4me3. These observations suggest that Fam60a is localized mostly to transcribed gene promoters, where it regulates the level of gene expression both negatively and positively via Sin3a and Tet.

How might Fam60a regulate Tet activity? It may inhibit dioxygenase enzymatic activity or impair recruitment of Tet to DNA. In this regard, PGC7 (also known as Stella) protects the female pronucleus from Tet3-dependent conversion of 5mC to 5hmC in mouse zygotes as well as inhibits the binding of Tet3 to chromatin in mouse ES cells (*Nakamura et al., 2012*). Fam60a may similarly affect the binding of Tet to chromatin, although this is unlikely given that recruitment of Tet1 to Fam60a target genes was not affected in *Fam60a^-/-^* ES cells. Alternatively, Fam60a may physically interact with Tet proteins and inhibit their activity. However, given that Tet proteins were not identified in our search for Fam60a-interacting proteins, it is unlikely that Fam60a directly interacts with Tet. Further characterization of the mechanisms by which Fam60a affects the function of Sin3a and Tet should provide new insight into gene regulation during embryogenesis.

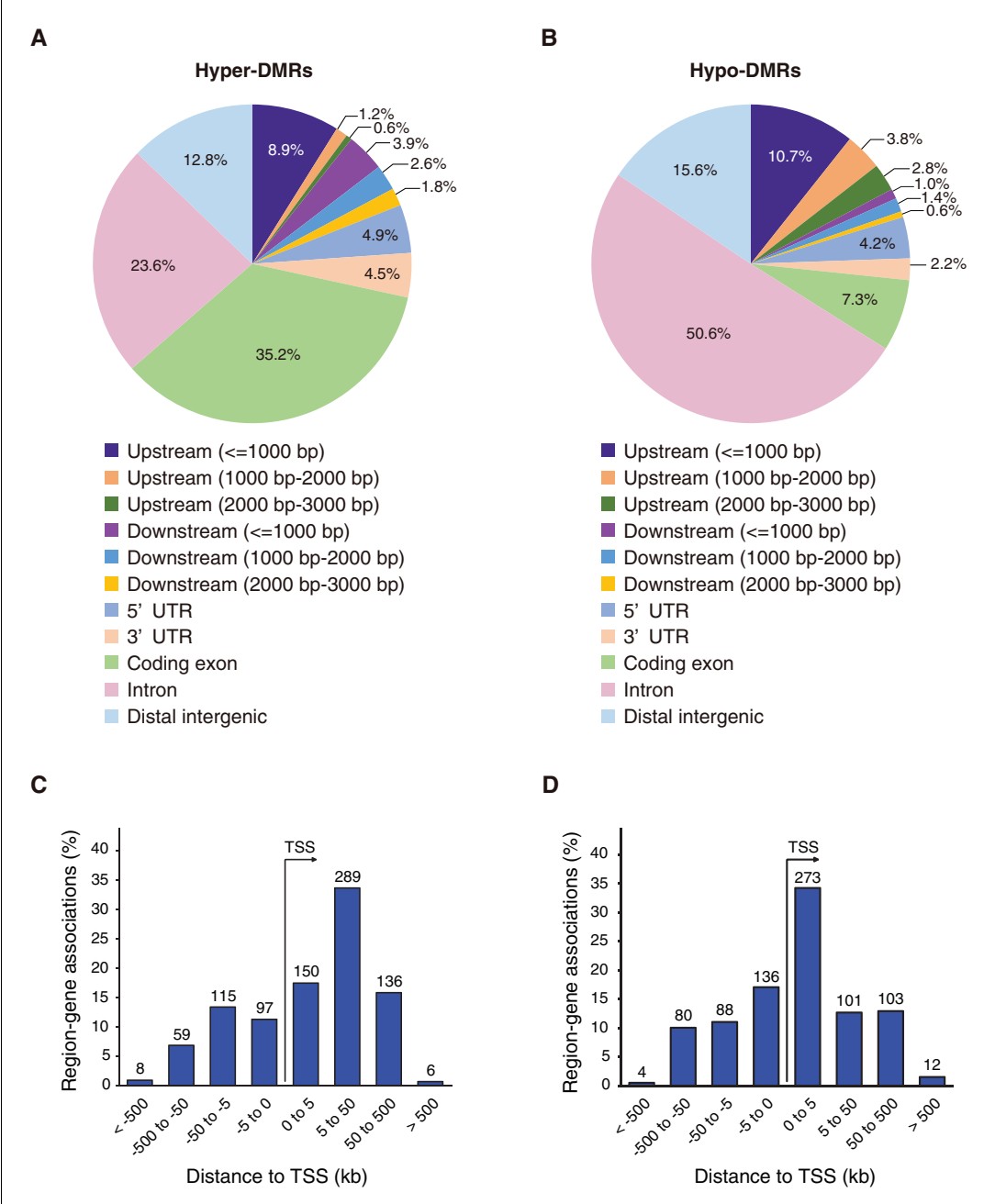

**Figure 8.** Differentially methylated regions (DMRs) in *Fam60a*[−/−] embryos. (**A** and **B**) Distribution of the top 500 hyper- and hypomethylated DMRs, respectively, among various genomic features. (**C** and **D**) Genomic position profile for the top 500 hyper- and hypomethylated DMRs, respectively, relative to the nearest TSSs. Note that the total number exceeds 500 because TSSs on both strands, in the vicinity of a DMR, are counted. See also *Figure 8—figure supplements 1* and *2* and *Supplementary file 2*.

DOI: https://doi.org/10.7554/eLife.36435.043

The following source data and figure supplements are available for figure 8:

**Source data 1.** List of hypo- and hypermethylated DMRs.
DOI: https://doi.org/10.7554/eLife.36435.047
**Figure supplement 1.** Genome-wide DNA methylation profiles of *Fam60a*[−/−] and WT embryos at E9.5.
DOI: https://doi.org/10.7554/eLife.36435.044
**Figure supplement 2.** Heatmaps with DNA methylation levels over Fam60a-bound promoters.
DOI: https://doi.org/10.7554/eLife.36435.045
**Figure supplement 3.** Average changes in DNA methylation in hyper- and hypomethylated DMRs.

*Figure 8 continued on next page*

*Figure 8 continued*

DOI: https://doi.org/10.7554/eLife.36435.046

# Materials and methods

## Key resources table

| Reagent type (species) or resource | Designation | Source or reference | Identifiers | Additional information |
|---|---|---|---|---|
| Gene (*mus musculus*) | fam60a | NA | NCBI Gene: 56306 | also known as SINHCAF |
| Gene (*mus musculus*) | Tet1 | NA | NCBI Gene: 52463 | |
| Strain, strain background (*mus musculus*) | ICR | charles river | | |
| Strain, strain background (*mus musculus*) | C57BL/6J | charles river | | |
| Strain, strain background (*mus musculus*) | 129 | charles river | | |
| Strain, strain background (*mus musculus*) | B6C3F1/Crl | charles river | | |
| Genetic reagent (EMCV) | internal ribosomal entry site (IRES)-βgeo | NA | | |
| Genetic reagent (P1 phage) | loxP | NA | | |
| Genetic reagent (P1 phage) | FRT | NA | | |
| Genetic reagent (*Saccharomyces cerevisiae*) | CAG-Flpe | PMID: 16651697 | | |
| Genetic reagent (P1 phage) | CAG-Cre | PMID: 9268708 | | |
| Genetic reagent (Aequorea victoria) | Fam60a-Venus | this paper | | |
| Genetic reagent (P1 phage) | Fam60a-CreERT2 | this paper | | |
| Cell line (*mus musculus*) | P19 | PMID:7056443 | | |
| Cell line (*mus musculus*) | NIH3T3 Tet-On 3G | Clontech | 631197 | |
| Antibody | Antibody to Fam60a (α-E15W) (rabbit polyclonal) | this paper | | 1/1000 dilution for IHC or WB |
| Antibody | anti-GFP (rabbit polyclonal) | MBL | Code No.598 RRID: AB_591819 | 10 µl for IP, 1/2000 dilution for IF |
| Antibody | control Rabbit IgG | Kamiya Biomedical | PC-124 | used for IP control |
| Antibody | control Rabbit IgG | Thermo Fisher Scientific | | used for IP control |
| Antibody | anti-HDAC1 (mouse monoclonal) | abcam | ab31263 RRID: AB_732774 | this product is discontinued by abcam |
| Antibody | anti-FLAG (mouse monoclonal) | Sigma-Aldrich | F3165 RRID: AB_259529 | 1/2000 for IF |
| Antibody | anti-HDAC2 (rabbit polyclonal) | abcam | ab7029 RRID: AB_305706 | 1/1000 dilution for WB |
| Antibody | anti-Sin3a (rabbit polyclonal) | Santa Cruz Biotechnology | sc-994 RRID: AB_2187760 | 1/1000 dilution for WB |
| Antibody | anti-Ing2 (rabbit polyclonal) | abcam | ab109504 RRID: AB_10861294 | 1/2000 dilution for WB |
| Antibody | anti-BrdU (mouse monoclonal) | BD bioscience | 347580 RRID: AB_10015219 | 1/200 dilution for IHC |

*Continued on next page*

Continued

| Reagent type (species) or resource | Designation | Source or reference | Identifiers | Additional information |
|---|---|---|---|---|
| Antibody | anti-5hmC (rabbit polyclonal) | active motif | 39769 RRID: AB_10013602 | 1/2000 dilution for IF |
| Antibody | anti-Histone H3K9ac (rabbit polyclonal) | active motif | 39917 RRID: AB_2616593 | used for ChIP assay |
| Antibody | anti-RbAp46/48 (rabbit polyclonal) | active motif | 39199 RRID: AB_2615007 | 1/2000 dilution for WB |
| Recombinant DNA reagent | pTRE3G-FLAG-Tet1 | this paper | | |
| Recombinant DNA reagent | pEF-BOS-Fam60a-IRES-Venus | this paper | | |
| Recombinant DNA reagent | pEF-BOS | PMID: 1698283 | | |
| Peptide, recombinant protein | E15W | this paper | | for the rise of Anti-Fam60a antibody |
| Peptide, recombinant protein | recombinant GFP protein | abcam | ab85191 | |
| Commercial assay or kit | EpiTect Bisulfite Kit | Qiagen | ID: 59104 | |
| Commercial assay or kit | PrimeScript RT Reagent Kit with gDNA Eraser | Takara | RR047A | |
| Commercial assay or kit | SOLiD Total RNA-Seq Kit | Life Technologies | 4445374 | |
| Commercial assay or kit | SureSelect Methyl-Seq Target Enrichment System | Agilent Technologies | 931052 | |
| Commercial assay or kit | EZ Methylation-Gold Kit | Zymo Research | | |
| Chemical compound, drug | BS3 | Thermo Fisher Scientific | Prod#21580 | for antibody conjugation to dynabeads |
| Chemical compound, drug | Doxycycline hyclate | Sigma-Aldrich | D9891 | |
| Chemical compound, drug | tamoxifen | Sigma-Aldrich | T5648-1G | dissolved in corn oil |
| Software, algorithm | LifeScope software | Applied Biosystem | | |
| Software, algorithm | MACS | PMID: 18798982 | | |
| Software, algorithm | CEAS | PMID: 19689956 | | |
| Software, algorithm | QUMA | PMID: 18487274 | | |
| Software, algorithm | bowtie2 | PMID: 22388286 | | |
| Software, algorithm | Bismark | PMID: 21493656 | | |
| Software, algorithm | Samtools | PMID: 19505943 | | |
| Software, algorithm | Picard toolkit | Broad Institute | | |
| Software, algorithm | methylKit program | PMID: 23034086 | | |
| Software, algorithm | BSseq program | PMID: 23034175 | | |
| Software, algorithm | bedtools | PMID: 20110278 | | |
| Software, algorithm | GREAT | PMID: 20436461 | | |
| Software, algorithm | aLeaves | PMID: 23677614 | | |
| Software, algorithm | MAFFT | PMID: 23329690 | | |
| Software, algorithm | trimAl | PMID: 19505945 | | |
| Software, algorithm | RAxML | PMID: 24451623 | | |

## Mice

$Fam60a^{\beta geo}$, a mutant allele of $Fam60a$ in which an internal ribosome entry site (IRES)–βgeo cassette and a loxP site are inserted in intron 4 and intron 1, respectively, was generated by gene targeting

in mouse ES cells (*Figure 2—figure supplement 1A*). A *Fam60a*flox allele was subsequently generated with the use of the *CAG-Flpe* transgene (*Kanki et al., 2006*), and a *Fam60a*− allele lacking exons 2 to 4 was generated with the use of the *CAG-Cre* transgene (*Sakai and Miyazaki, 1997*). Both *Fam60a*βgeo and *Fam60a*− alleles are functionally null. Mutant mice were maintained on the 129/C57B6 mixed background. PCR primers for genotyping were Fam60a-5A (5′-ATATGCTGC TAGGTGCCACAG-3′), Fam60a-3A (5′-TTCTCTACTCCATAGCACAGG-3′), and Fam60a-3C (5′-CTAC TGTGGTCACAAGCAGAC-3′). A BAC transgene (*Fam60a::Venus*) encoding a Fam60a-Venus fusion protein was constructed from mouse BAC clone RP23-100A22 with the use of a BAC recombination system (*Figure 3—figure supplement 1A*) (*Copeland et al., 2001*). The Fam60a-Venus protein, in which Venus is fused to the COOH-terminus of Fam60a, is functional, given that the transgene is able to rescue the phenotype of *Fam60a* mutant mice (*Figure 3—figure supplement 1C*). A BAC transgene (*Fam60a-CreERT2*) was constructed by inserting CreERT2 into the *Fam60a* BAC clone.

## Cell line origin and authentication

P19 embryonal carcinoma cell line (*McBurney and Rogers, 1982*) is a gift from Michael McBurney (University of Ottawa). NIH3T3 Tet-On 3 G cell line (631197, Clontech) was purchased from Clontech, Takara-bio (Kyoto, Japan).

## Identification of Fam60a-interacting proteins

E10.5 embryos harboring the *Fam60a::Venus* transgene were recovered in PBS for the preparation of nuclear extracts. The embryos were passed through a 70 µm cell strainer with a plunger, and the cells were allowed to swell by incubation in buffer A (10 mM Hepes-KOH (pH 7.9), 10 mM KCl, 1.5 mM MgCl$_2$, 0.1 mM EGTA, 1 mM dithiothreitol, and Roche complete protease inhibitor cocktail) for 15 min on ice before homogenization with 20 strokes of a loose-fitting pestle in a Dounce homogenizer. Nonidet P-40 was then added to the homogenate at a final concentration of 0.1%, and another 20 strokes of the pestle were applied. The homogenate was centrifuged at 960 × *g* for 5 min at 4°C, and the resulting nuclear pellet was suspended and incubated for 3 hr at 4°C either in RIPA buffer (50 mM Tris-HCl (pH 8.0), 150 mM NaCl, 2 mM EDTA, 1% Nonidet P-40, 0.5% sodium deoxycholate, 0.1% SDS, 1 mM dithiothreitol, and Roche complete protease inhibitor cocktail), in buffer C (20 mM Hepes-KOH (pH 7.9), 400 mM NaCl, 0.1 mM EDTA, 0.1 mM EGTA, 0.1% Nonidet P-40, 1 mM dithiothreitol, and Roche complete protease inhibitor cocktail), or in nondenaturing lysis buffer containing Benzonase nuclease (20 mM Tris-HCl (pH 8.0), 137 mM NaCl, 2 mM EGTA, 1.5 mM MgCl$_2$, 10% glycerol, 1 mM dithiothreitol, Benzonase nuclease (125 U; 70,446–3, Novagen), and Roche complete protease inhibitor cocktail). The samples were centrifuged at 18,000 × *g* for 10 min at 4°C, and the resulting supernatants (nuclear extracts) were incubated with Dynal Protein G beads (Invitrogen) for 3 hr at 4°C. After removal of the beads, the extracts were divided into two halves. One half was incubated for 3 hr at 4°C with Dynal Protein G beads conjugated with antibodies to GFP, whereas the other half was incubated with identical antibody-conjugated beads that had been previously exposed to recombinant GFP (ab84191, Abcam) to mask the antigen binding site. Proteins that bound to the beads were eluted by incubation for 30 min at 37°C with 1 × SDS sample buffer not containing dithiothreitol. They were then fractionated by SDS-polyacrylamide gel electrophoresis and silver-stained. Target proteins were identified by liquid chromatography and tandem mass spectrometry with a nano-UPLC Q-TOF MS/MS system (SYNAPT G2, Waters).

## Immunoprecipitation and immunoblot analysis

Nuclear extracts prepared from E10.5 embryos or undifferentiated P19 cells with RIPA buffer as described above were incubated for 3 hr at 4°C first with Dynal Protein G beads alone and then with antibody-conjugated beads. Proteins that bound to the antibody-conjugated beads were eluted by incubation for 30 min at 37°C with 1 × SDS sample buffer not containing dithiothreitol, fractionated by SDS-polyacrylamide gel electrophoresis, and transferred to a polyvinylidene difluoride membrane. The membrane was then subjected to immunoblot analysis with primary antibodies, horseradish peroxidase–conjugated secondary antibodies, and ECL Plus reagents (RPN2133, Amersham).

## Lineage tracing

Tamoxifen (6 mg; T5648, Sigma-Aldrich) in 1 ml of corn oil (C8267, Sigma-Aldrich) was administered orally to *Fam60a-CreERT2::ROSA26RlacZ* mice at the age of 8 weeks age. One, 3, or 5 days after tamoxifen administration, mice were killed and the duodenum was removed and then fixed overnight at 4°C in phosphate-buffered saline (PBS) containing 1% paraformaldehyde, 0.2% glutaraldehyde, and 0.02% Nonidet P-40. Expression of the *lacZ* transgene was detected by staining with X-gal as described previously (*Saijoh et al., 1999*).

## In situ hybridization and histology

Embryos were dissected in PBS and fixed with 4% paraformaldehyde. In situ hybridization was performed with whole-mount preparations (*Sakai et al., 2001*) or sections (*Yashiro et al., 2000*). The 3′untranslated region of *Fam60a* was used as a probe for in situ hybridization. For histological analysis, embryos were fixed with 4% paraformaldehyde, dehydrated, and embedded in paraffin. Serial sections (thickness, 7 µm) were stained with hematoxylin-eosin according to standard procedures.

## Antibodies

Antibodies to Fam60a (α-E15W) were generated in rabbits by injection of a synthetic peptide corresponding to the COOH-terminal region of the mouse protein (EEQGPAPLPISTQEW) and were affinity-purified. Additional antibodies included control rabbit IgG (Kamiya Biomedical or Thermo Fisher Scientific), conformation-specific mouse monoclonal antibodies to rabbit IgG (#3678, Cell Signaling) that can avoid detection of denatured rabbit IgG used for immunoprecipitation, as well as rabbit polyclonal antibodies to GFP (598, MBL International), to Hdac1 (ab31263, Abcam), to FLAG (F3165, Sigma-Aldrich), to Hdac2 (ab7029, Abcam), to Sin3a (sc-994, Santa Cruz Biotechnology), to Ing2 (ab109504, Abcam), to RbAp46/48 (39199, Active Motif), to AcH3K9 (39917, Active Motif), to BrdU (347580, BD Biosciences), and to 5hmC (39769, Active Motif). Mouse monoclonal antibodies to FLAG for immunostaining were obtained from Sigma.

## Immunostaining of embryos

Cryosections were incubated overnight at 4°C with primary antibodies. Immune complexes were detected with horseradish peroxidase–conjugated secondary antibodies (ImmPRESS reagent, Vector labs) or Alexa Fluor 488–conjugated secondary antibodies (Molecular Probes). Nuclei were counterstained with 4′,6-diamidino-2-phenylindole (DAPI). Confocal images were acquired with a confocal microscope (Olympus FV1000D or Zeiss LSM510META).

## BrdU incorporation assay

Pregnant mice were injected intraperitoneally with undiluted BrdU labeling reagent (RPN20LR, Amersham) at a dose of 1 ml per 100 g of body weight. Embryos were dissected 30 min after BrdU injection and were subjected to immunostaining of BrdU as previously described (*Santarelli et al., 2003*) with the use of a Vectastain ABC Kit (Vector labs) and diaminobenzidine. The proliferation index was calculated as the percentage of cells positive for BrdU incorporation.

## RT-qPCR analysis

Total RNA was isolated from E9.5 embryos with the use of TRIzol reagent (15596026, Invitrogen), and portions of the RNA (1 in 20 µl) were subjected to RT with the use of a PrimeScript RT Reagent Kit with gDNA Eraser (RR047A, Takara). The resulting cDNA (corresponding to an RNA amount of 15, or 0.92 ng for β-actin qPCR) was subjected to real-time PCR analysis with the use of Power SYBR Green PCR Master Mix (4367659, Applied Biosystems). For quantitation of mRNAs, we established standard curves with serial dilutions of RNA of known concentrations. Data were normalized byβ-actin mRNA abundance. PCR primers (forward and reverse, respectively) were as follows: 5′-GGTCATCACTATTGGCAACG-3′ and 5′-ACGGATGTCAACGTCACACT-3′ for *Actb* (β-actin); 5′-TACCAGGGTAGCAACCCAAT-3′ and 5′-GGTTTCTGACAGCCCTCTTC-3′ for *Adhfe1*; 5′-CTGATTGAGGAGTTGAGGC-3′ and 5′-AGCCTACAGTTGGAGCCTG-3′ for *Nagk*; 5′-CTGACTTTTCGGTGGGCTACA-3′ and 5′-GGCGCAGAATGGCTCTTC-3′ for *Leng9*; 5′-GGCCTGCCTCCTTTAGTCTC-3′ and 5′-TGTCAGCATCTGTGGCTGTT-3′ for *Dchs1*; 5′-GCTCAGACTCAGACCAAGAG-3′ and 5′-TGCTGTGTGAGTAGCTGTGC-3′ for *Mxd3*; 5′-GGGGGAATGAGTGCTTGAAG-3′ and 5′-TCACCTGGACC

TCCAAATGTC-3′ for *AA465934*; 5′-AAAGAGAAACTGCCAACGC-3′ and 5′-TATTCATACC TGGGCCGAAG-3′ for *2610020C07Rik*; 5′-GTAGGGGATCGGGACTCTGG-3′ and 5′-TCCTCAAG- GAATGATCCGGC-3′ for *Gt(ROSA)26Sor*; and 5′-CAATTACAGAAGTGTGGGACT-3′ and 5′-CACC TTCCTCCCAGTTCTTT-3′ for *Acsl3*.

## ChIP-seq and RNA-seq

E9.5 embryos harboring the *Fam60::Venus* transgene were recovered in PBS for ChIP with antibodies to GFP performed as previously described (*Hayakawa et al., 2007*). The isolated DNA was applied to ChIP-seq library construction with the use of a SOLiD Fragment Library Core Kit (PN 4464412, Life Technologies). Sequencing was performed with a SOLiD four instrument (Life Technologies). Sequenced reads were aligned to the mouse genome (mm9) with the use of LifeScope software (Applied Biosystems). Aligned peaks were called and BED and Wig files were generated with MACS version 1.4.1 (*Zhang et al., 2008*), and the files were visualized in the UCSC genome browser as custom tracks. The called peaks were filtered with the following criteria: false discovery rate (FDR) of ≤1% and fold enrichment of ≥2.0. To obtain a peak distribution and averaged peak profile around genes, we analyzed the filtered peaks with CEAS version 1.0.2. Genes with filtered peaks within ±3 kb of the TSS in UCSC RefGene were defined as Fam60a target genes.

For RNA-seq, E9.5 embryos were collected in PBS and stored in RNAlater (AM7020, Ambion) at –80°C. After genotyping with yolk sac DNA, RNA was isolated from WT and *Fam60a*$^{-/-}$ embryos with the TRIzol reagent and mRNA was extracted twice with the use of a MicroPoly(A) Purist Kit (AM1922, Life Technologies). Library preparation was performed with the use of a SOLiD Total RNA-Seq Kit (4445374, Life Technologies). Three biological replicates were analyzed for each genotype. Libraries were labeled with distinct barcoding adapters. Sequencing was performed with a SOLiD4 instrument, and sequencing data were mapped to the mouse genome (mm9) with the use of Life-Scope software. Differentially expressed genes were identified with the edgeR Bioconductor package. Transcripts with an FDR of <0.01 were considered to be significantly up- or down-regulated.

## Bisulfite sequencing

Genomic DNA was isolated from WT and *Fam60a*$^{-/-}$ embryos according to standard procedures, and its concentration was determined by spectrophotometry. The DNA (500 to 1000 ng) was treated with bisulfite and purified with the use of an EpiTect Bisulfite Kit (59104, Qiagen) and was then subjected to PCR amplification with the following primer sets: 5′- ATTTAGTGGGGTTTTTGTTATTG-3′ (Adhfe1 Bis F1) and 5′-TATTTCTACACATAAACCCATAC-3′ (Adhfe1 Bis R1) for initial PCR and Adhfe1 Bis F1 and 5′-ACTAAACCACATTACACCATCC-3′ (Adhfe1 Bis R2) for seminested PCR; 5′- TGGAAGGAGGTTAAAGGATTAG-3′ (Leng9 Bis F1) and 5′-AAATTATCTAAACCCTACCCCC-3′ (Leng9 Bis R1); 5′-ATTTTTTTAGGAGTTTTAGTTGGGGTG-3′ (Nagk Bis F1) and 5′-CAACTCTACA- CAACTCTCCAAATTAAC-3′ (Nagk Bis R1); 5′-AGAGGGTGTATGTTGTAGAGTAGTTAGGTG-3′ (Peg3 Met11) and 5′-CATCCCATCCCCCTTTTCCAAACTCTAC-3′ (Peg3 Met12.1); and 5′-GTA TTTAGTTTATTATGAGGAAGAGTTT-3′ (Kcnq1ot1 1F) and 5′-CAAAAACAACTCCAAAAAAACTA TAAA-3′ (Kcnq1ot1 1R). The amplified fragments were separated by agarose gel electrophoresis and the target bands excised. DNA was recovered from the excised gel pieces with the use of a QIAquick Gel Extraction Kit (28706, Qiagen) and was then cloned into the pCRII vector with the use of a Dual Promoter TA Cloning Kit (K207020, Invitrogen). Sequenced fragments were analyzed with the QUMA tool (quantification tool for methylation analysis; http://quma.cdb.riken.jp).

## Forced expression of Fam60a and FLAG-Tet1

NIH3T3 Tet-On 3G fibroblasts (631197, Clontech) were seeded at ~80% confluence on 15-mm-diameter cover slips coated with 0.1% gelatin and placed in 24-well plates. The cells were cultured for at least 2 hr at 37°C in Dulbecco's modified Eagle's medium supplemented with 10% fetal bovine serum and were then transfected for 24 hr with 125 ng of pTRE3G-FLAG-Tet1 (encoding FLAG-tagged mouse Tet1) with or without 250 ng of pEF-BOS-Fam60a-IRES-Venus (encoding mouse Fam60a and Venus) with the use of the Lipofectamine LTX reagent (15338500, Invitrogen). The cells transfected without or with pEF-BOS-Fam60a-IRES-Venus were also transfected with 375 or 125 ng, respectively, of the pEF-BOS empty vector. Expression of FLAG-Tet1 was induced by exposure of the cells to doxycycline (1 μg/ml) for 24 hr, after which the cells were fixed for 15 min with 4%

paraformaldehyde in PBS, permeabilized for 15 min with 0.2% Triton X-100 in PBS, treated for 20 min with 2 M HCl, neutralized for 10 min with 100 mM Tris-HCl (pH 8.0), washed with PBS, and exposed for 1 hr to blocking buffer (1% bovine serum albumin and 0.1% Tween 20 in PBS), all at room temperature. The cells were then incubated overnight at 4°C with mouse monoclonal antibodies to FLAG (1:2000 dilution) and rabbit polyclonal antibodies to 5hmC (1:2000 dilution) in blocking buffer. Immune complexes were detected with Alexa Fluor 568– or Alexa Fluor 647–conjugated secondary antibodies (Molecular Probes), respectively, and nuclei were stained with DAPI (250 ng/ml). The cells were mounted in ProLong Gold antifade reagent (P36930, Invitrogen), and images were acquired with a confocal microscope (Olympus FV1000D). The fluorescence intensity of 5hmC was plotted against that of FLAG. If FLAG fluorescence intensity was >40, the cell was considered as FLAG-Tet1 positive; if 5hmC fluorescence intensity was >30, the cell was considered as 5hmC positive.

## ChIP-qPCR

ChIP was performed as described above, and the precipitated DNA was subjected to qPCR analysis with the following primers (forward and reverse, respectively): 5′-CTAGCCACGAGAGAGCGAAG-3′ and 5′-AGCTTCTTTGCAGCTCCTTC-3′ for *Actb*; 5′-GACCGGATTGGCTGTTAGTG-3′ and 5′-TAGGTGCCTCAGCAAGTGTG-3′ for *Adhfe1*; 5′-CTAGGAAGAAGCGGCAGACC-3′ and 5′-GGCGTCACAGTTGGAGATCA-3′ for *Leng9*; 5′-CTGAGATTCATGCACAAGGG-3′ and 5′-TATAGGAACCAAGGGCGTTC-3′ for *Nagk*; 5′-GCGAGGACACTCACTGACTC-3′ and 5′-AGTGTGTGGTGGTGCTTGAG-3′ for *Dchs1*; 5′-GTGACGACAACTCGCGTAC-3′ and 5′-AATGGCCCTAATGAGAGACG-3′ for *Mxd3*; 5′-TTGGGAATCCAGTGGAAACT-3′ and 5′-AGCCATGCACAAAGTTCTTG-3′ for *Acsl3*; 5′-CTGGAGTTGCAGATCACGAG-3′ and 5′-CCTTTCTGGGAGTTCTCTGC-3′ for *Gt(ROSA)26Sor*; 5′-TAAAGAGAAACTGCCAACGC-3′ and 5′ CTCATAGGACGTTCTGGCG 3′ for *2610020C07Rik*; and 5′-CTGTCCAAGACTGCGGAATG-3′ and 5′-CCTGAAGCCATCCTTGGTAG-3′ for *AA465934*.

## hMeDIP analysis

E9.5 embryos were recovered in PBS and stored at –80°C. After genotyping, embryos were lysed overnight at 55°C in a solution containing 20 mM Tris-HCl (pH 8.0),

4 mM EDTA, 20 mM NaCl, 1% SDS, and proteinase K (0.4 mg/ml, Nacalai). They were then exposed for 30 min at 37°C to RNase A (5 mg/ml, Sigma) before purification of genomic DNA first by phenol-chloroform treatment and ethanol precipitation and then with the use of a QIAamp DNA Micro Kit (56304, Qiagen). The DNA was sheared with the use of a Bioruptor UCD-250 (Diagenode) (15 s on and 15 s off for 10 min at low power). Portions (500 ng) of the sheared DNA were denatured for 10 min at 98°C, placed on ice, and then incubated overnight at 4°C with rotation in 100 μl of hMeDIP buffer containing 20 mM Tris-HCl (pH 8.0), 2 mM EDTA, 150 mM NaCl, 1% Triton X-100, 4 μg of antibodies to 5hmC, and 1% bovine serum albumin. Dynal Protein G beads were then added to the samples to precipitate the antibody-DNA complexes, after which the beads were washed three times with hMeDIP wash buffer (20 mM Tris-HCl (pH 8.0), 2 mM EDTA, 300 mM NaCl, 1% Triton X-100, 0.1% SDS) and then treated overnight at 55°C with proteinase K in hMeDIP elution buffer (20 mM Tris-HCl (pH 8.0), 8 mM EDTA, 300 mM NaCl, 0.5% SDS). The eluted DNA was purified with the use of a QIAquick PCR Purification Kit (28106, Qiagen) and subjected to qPCR analysis with the primers described above for ChIP-qPCR.

## Methyl-seq library construction

Libraries compatible with the Illumina platform were prepared from 3 μg of genomic DNA with the use of a SureSelect Methyl-Seq Target Enrichment System (Agilent Technologies). Genomic DNA was sheared at 4°C by focused ultrasonic disruption with a Focused-ultrasonicator E220 (Covaris) (duty factor, 10%; PIP, 175; cycles per burst, 200; time, 360 s). The fragmented DNA was end-repaired, adenylated at the 3′ end, and ligated to a methylated adapter. The prepared libraries were subjected to hybridization with the biotinylated SureSelect Methyl-Seq Capture Library (Agilent Technologies), which covers genomic regions of 109 Mb in total including GENCODE promoters; CpG islands, shores, and shelves; DNase I–hypersensitive sites; and RefGenes. Library molecules that overlapped the targeted regions were collected with streptavidin-conjugated beads and converted with bisulfite with the use of an EZ Methylation-Gold Kit (Zymo Research) before amplification

by PCR. Further amplification was performed with the use of the SureSelect Methyl-Seq Indexing Primer (Agilent Technologies) to allow multiplexed sequencing on the Illumina platform.

## Methyl-seq and detection of DMRs

The amplified libraries supplemented with 20% of a phiX sequencing control library were sequenced with an Illumina HiSeq 1500 instrument with $2 \times 127$ cycles in the Rapid Run Mode. Sequence reads were obtained with HiSeq Control Software (HCS) version 2.2.58 and Real-Time Analysis (RTA) version 1.18.64.0. The obtained paired-end reads were subjected to quality control with FastQC version 0.11.5 (https://www.bioinformatics.babraham.ac.uk/projects/fastqc/), and adapter sequences and low-quality reads were removed using Trim Galore! version 0.4.2 and with the parameters '-e 0.1 -q 30' (http://www.bioinformatics.babraham.ac.uk/projects/trim_galore). After the removal of phiX-derived reads with Bowtie2 version 2.3.0 (*Langmead and Salzberg, 2012*), the valid reads were mapped to the UCSC mm9 reference genome sequence using Bismark version 0.17.0 (*Krueger and Andrews, 2011*) and with the parameters '–bowtie2 -N 1 L 22 –score_min L,-0.6,-0.6.' Before methylation calling at each CpG site, potential PCR duplicates were removed and only read-pairs from the expected strand (the original bottom strand of the reference genome sequence) were extracted with the use of Bismark and Samtools version 1.3.1 (*Li et al., 2009*), respectively. The on-bait coverage of mapped reads was calculated with CollectHsMetrics of the Picard package version 2.8.1 (http://broadinstitute.github.io/picard). Methylated CpG was identified using the bismark_methylation_extractor function of Bismark and with the parameter '–cutoff 5.' To compare methylation profiles among libraries, we performed a hierarchical clustering analysis according to Ward's method with the use of the methylKit program version 1.0.0 (*Akalin et al., 2012*) in the Bioconductor package. For detection of DMRs in three mutant embryos compared with three WT embryos, we used BSseq version 1.10.0 (*Hansen et al., 2012*) in the Bioconductor package. After importation of the CpG report files of the Bismark output, the BSseq data were processed with the BSmooth algorithm for computation of smoothed methylation levels. The smoothed methylation data were selected for regions with a read coverage of at least five reads at the CpG sites in at least two of the three samples in both comparison groups. Comparison of the mutant and WT samples was then performed with t-statistics. DMRs were detected on the basis of the threshold 'qcutoff (low = 0.025, high = 0.975)' and were further narrowed down to those with a minimum of three CpG sites and mean methylation difference of ≥0.05. For examination of the relation between DMRs and Fam60a ChIP-seq peak regions, the peaks of the two ChIP-seq analyses were merged on the basis of their genomic locations and the merged peaks were then compared with DMRs with the use of bedtools version 2.26.0 (http://bedtools.readthedocs.io). Regions of overlap were characterized by statistical evaluation of peak enrichment at genome features such as promoters, exons, introns, untranslated regions (UTRs), and distal intergenic regions with the use of CEAS version 0.9.9.7 (*Shin et al., 2009*), and plots of average profiles near TSSs were constructed with GREAT version 3.0.0 (*McLean et al., 2010*). Methylation levels at imprinted genes and DMRs were visualized with the UCSC Integrative Genomics Viewer (IGV) version 2.3.72 (*Thorvaldsdóttir et al., 2013*).

## Molecular phylogenetics

Amino acid sequences similar to that of human Fam60a were collected by aLeaves (*Kuraku et al., 2013*), and the resultant sequence set was then modified to remove redundant sequences. The modified sequence set was subjected first to multiple alignment with the use of the program MAFFT v7.299b (*Katoh and Standley, 2013*) and with the option '-linsi' and then to trimming of unaligned and gapped sites with the program trimAl v1.4.rev15 (*Capella-Gutiérrez et al., 2009*) with the options '-automated1' and '-nogaps' in order. The obtained sequence file was used to infer the maximum-likelihood tree with the program RAxML v8.2.8 (*Stamatakis, 2014*) according to the PROT-CATWAG model and with 1000 bootstrap resamplings.

## Data availability

RNA-seq, ChIP-seq and Methyl-seq data have been deposited in DNA Data Bank of Japan (DDBJ) with the accession numbers DRA004841, DRA004842 and DRA006579, respectively.

## Statistical analysis

Quantitative data are presented as means ± s.d. and were analyzed with the unpaired Student's *t* test. A p value of $< 0.05$ was considered statistically significant.

## Acknowledgements

We thank Shinsuke Ito for Tet expression vectors and discussion; Hiroshi Kimura and Shunsuke Toyoda for discussion; Kaori Tatsumi and other members of the Laboratory for Phyloinformatics at the RIKEN Center for Biosystems Dynamics Research for generation of Methyl-seq data; and Akemi Fukumoto, Yayoi Ikawa, and Hiromi Nishimura for technical assistance. This study was supported by a grant from the Ministry of Education, Culture, Sports, Science, and Technology of Japan (25251029).

## Additional information

### Funding

| Funder | Grant reference number | Author |
| --- | --- | --- |
| Ministry of Education, Culture, Sports, Science, and Technology | 25251029 | Hiroshi Hamada |

The funders had no role in study design, data collection and interpretation, or the decision to submit the work for publication.

### Author contributions

Ryo Nabeshima, Conceptualization, Supervision, Funding acquisition, Project administration, Writing—review and editing; Osamu Nishimura, Conceptualization, Formal analysis, Investigation, Methodology, Writing—original draft; Takako Maeda, Data curation, Software, Formal analysis, Investigation, Methodology; Natsumi Shimizu, Yasuo Sakai, Formal analysis, Investigation; Takahiro Ide, Hidetaka Shiratori, Software, Formal analysis, Investigation; Kenta Yashiro, Formal analysis, Investigation, Methodology; Chikara Meno, Mitsutaka Kadota, Resources, Investigation; Shigehiro Kuraku, Investigation, Methodology; Hiroshi Hamada, Conceptualization, Formal analysis, Investigation, Project administration, Writing—review and editing

### Author ORCIDs

Ryo Nabeshima http://orcid.org/0000-0002-5096-8306
Osamu Nishimura http://orcid.org/0000-0003-1969-2580
Natsumi Shimizu http://orcid.org/0000-0003-2671-4813
Takahiro Ide http://orcid.org/0000-0003-0709-4057
Shigehiro Kuraku http://orcid.org/0000-0003-1464-8388
Hiroshi Hamada http://orcid.org/0000-0002-7196-5948

### Ethics

Animal experimentation: All mouse experiments were approved by the relevant committees of Osaka University and RIKEN Center for Developmental Biology, license numbers FBS-12-019 and AH28-01

### Decision letter and Author response

Decision letter https://doi.org/10.7554/eLife.36435.058
Author response https://doi.org/10.7554/eLife.36435.059

## Additional files

### Supplementary files

• Supplementary file 1. Distribution of *Fam60a* genotypes for mouse embryos obtained by heterozygote intercrosses at various stages of development.
DOI: https://doi.org/10.7554/eLife.36435.048

• Supplementary file 2. Methylation level of CpG sites in the captured DNA in three embryos of each genotype.
DOI: https://doi.org/10.7554/eLife.36435.049

• Transparent reporting form
DOI: https://doi.org/10.7554/eLife.36435.050

### Data availability

RNA-seq, ChIP-seq and Methyl-seq data have been deposited in DNA Data Bank of Japan (DDBJ) with the accession numbers DRA004841, DRA004842 and DRA006579, respectively.

The following datasets were generated:

| Author(s) | Year | Dataset title | Dataset URL | Database, license, and accessibility information |
|---|---|---|---|---|
| Nabeshima R, Hamada H | 2016 | Fam60a is a component of the mSin3A-HDAC transcriptional corepressor complex and inhibits Tet-mediated DNA demethylation | www.ncbi.nlm.nih.gov/bioproject/?term=PRJDB4950 | Publicly available at NCBI BioProject (accession no: PRJDB4950) |
| Nabeshima R, Hamada H | 2016 | Fam60a is a component of the mSin3A-HDAC transcriptional corepressor complex and inhibits Tet-mediated DNA demethylation | www.ncbi.nlm.nih.gov/bioproject/?term=PRJDB4951 | Publicly available at NCBI BioProject (accession no: PRJDB4951) |
| Nabeshima R, Nishimura O, Kadota M, Shimizu N, Kuraku S, Hamada H | 2018 | Fam60a is a component of the mSin3A-HDAC transcriptional corepressor complex and inhibits Tet-mediated DNA demethylation | www.ncbi.nlm.nih.gov/bioproject/?term=PRJDB6732 | Publicly available at NCBI BioProject (accession no: PRJDB6732) |

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
