## [Decision Letter]

[Editors’ note: a previous version of this study was rejected after peer review, but the authors submitted for reconsideration. The first decision letter after peer review is shown below.]

Thank you for submitting your work entitled "Fam60a is a component of the mSin3a-Hdac transcriptional corepressor complex and inhibits Tet-mediated DNA demethylation" for consideration by *eLife*. Your article has been reviewed by three peer reviewers, and the evaluation has been overseen by a Reviewing Editor and a Senior Editor. The reviewers have opted to remain anonymous.

Our decision has been reached after consultation between the reviewers. Based on these discussions and the individual reviews below, we regret to inform you that your work will not be considered for immediate publication in *eLife*.

The major concern is that the regulation of DNA methylation and Tet1 activity by Fam60a is not sufficiently well established. Furthermore, the different sections of the manuscript – developmental functions of Fam60, biochemistry and links with the Sin3a complex and the genomics – do not connect well and read almost as three different manuscripts.

Given that responding to the reviewers' comments is likely to take more than the two month period allowed for revisions, we think it best to formally reject the paper but would encourage re-submission once the concerns have been addressed. Although this will be considered a new manuscript, we would make every effort to return the paper to the original reviewers.

*Reviewer #1:*

The manuscript entitled "Fam60a is a component of the mSin3a-Hdac transcriptional corepressor complex and inhibits Tet-mediated DNA demethylation" shows that Fam60a modulates DNA methylation, possibly by modulating Tet activity. The impact of this finding is that no function has been previously associated with Fam60a, and its possible association with methylation in stem cells is significant.

Having previously found differential expression of Fam60a in ESCs and differentiated cells, the authors examined expression throughout development and in adults to find that expression correlated with somatic stem cells in the intestinal crypt. They developed a *Fam60a-CreERT2* transgene to examine lineage expression in crypt cells, and found expression in the villi that migrated up to the tip. They then developed a knockout to find that embryos die in mid-gestation, and have general organ hypoplasia as well as cardiac outflow tract defects. They developed a *Fam60a::Venus* BAC transgenic mouse to identify interacting proteins using IP-mass spec. Interestingly, mSin3A and HDAC1 were found as strong interacting partners. They carried out ChIP-seq to find that Fam60a is recruited to promoter regions of genes to regulate gene expression, and compared these data with RNASeq. The authors show convincing evidence that 5-hydroxymethylation, a product of Tet activity, increases when Fam60a is absent in transfected NIH3T3 cells, yet do not provide direct evidence for how Fam60a might be doing so. A further concern is that upon comparing mutant and wild type embryos for differential methylation at putative Fam60a target genes, only one gene appears to be differentially methylated. Therefore, how differential methylation by Tet is modulated by Fam60a remains obscure.

The manuscript reads as if it is two different papers. One story touches on the role of Fam60a in development and intestinal stem cells, and the second assesses its biochemical function. This is somewhat of a problem for genes for which no function is known, but I feel that the manuscript would be more appealing to a wider audience if it could be tightened up somewhat to focus on the role of Fam60a in stem cells and methylation. The story should also be linked again in the Discussion.

The embryo expression at E9.5 would be very difficult for anyone who is not an expert to interpret, and could be placed in supplementary, focusing on intestinal expression. The lineage tracing experiment in intestinal villi is very nice, but loses relevance in Figure 1. Further, arrows pointing to key regions would help to guide the reader. It is not clear if the expression data correlate with the heart defects seen in the knockout, or if the KO mice have intestinal defects.

Some data are confusing and do not support the general claims of the authors. For example, ChIPSeq identified a large number of putative gene targets, yet differential methylation was found at only one – *Adhfe1*. Without definition of these genes, interpretation of any biology is difficult. Sin3a-Hdac would normally alter AcH3K9, yet this mark did not change at Fam60a target genes in WT and mutant embryos. Some discussion of these unexpected data would be warranted.

*Reviewer #2:*

The manuscript by Nabeshima et al. explores the role of Fam60a, a protein highly expressed in ESCs. The first part of the manuscript describes the phenotype of the mouse *Fam60* mutant and the protein interactions of Fam60 including the Sin3A complex components. This is followed by genomic analyses of *Fam60* binding and the transcriptome analyses of the mutant. The authors then attempt to build a link between Fam60 and Tet1 in cultured cells and mouse embryos. In my opinion, the majority of the genomic data appears very preliminary and the presented findings frequently do not support the drawn conclusions. The links between Fam60 action and Tet1 are especially weak and built mostly on conjecture. I also found the quality of the figures (especially the figures dealing with ChIP-seq and RNA-seq data) substandard while many other figures are not properly labelled and thus confusing. The biochemical data are of some interest due to the broad spectrum of Sin3 functions. Unfortunately however, those experiments were not followed up on. My comments and suggestions for improvement are outlined below.

What is the reason for including the MA plot in Figure 4A? It comes with very little/no explanation and if its only purpose is to show that *Fam60* loss results in bidirectional gene changes, then it is redundant with the B panel.

Did the authors assess whether the upregulated and downregulated genes are enriched in any specific GO categories? Some gene network analyses would also be very beneficial since there are currently no strong links between the mouse phenotypes and the RNA-seq data.

Why are there no error bars in Figure 4C? I assume that this data corresponds to the average of the three RNA-seq experiments.

I find it very problematic that the *Fam60* ChIP-seq experiment has no sequenced input controls (at least I couldn't find any). This can result in suboptimal peak calling and biased downstream analyses. For example, the observed enrichment of Fam60 peaks around the TSS could be partly due to the GC content sequencing bias.

What is the reason for the double peak in Figure 5A?

What does "+/- 3Kb" means in Figure 5C? Were the peaks extended for 3Kb in each direction before the overlaps were performed?

Subsection “Fam60a inhibits Tet1 activity in cultured cells”: A Table is not an adequate tool for evolutionary comparisons. To properly address this issue, the authors should construct phylogenetic trees for DNMT proteins, Tet proteins, Sin3 and Fam60a. Also, how did the authors decide which organisms to include in this comparison? I would suggest including basal metazoans (such as Cnidaria and Ctenophora that are known to have DNA methylation) as well as cephalochordates and tunicates. A few insects known to employ DNA methylation (such as ants and wasps) can also be included.

Figure 6: The data presented in Figure 6 is not very convincing. The effect of *Fam60* on Tet1 expression is not very robust and the statistical significance of the presented differences appears to be low. Also, I find the system that the authors use highly artificial. A better experiment would be to generate *Fam60a^-/-^* cells (for example ESCs that express both proteins) and perform a Tet1 ChIP-seq to assess whether the loss of Fam60 results in increased binding of Tet1. This experiment could also be complemented with base-resolution hmC profiling.

Subsection “Aberrant promoter hypomethylation in *Fam60a^–/–^* mouse embryos” and Figure 7: It is very difficult to make any definite conclusions regarding the relationships between Tet1 and *Fam60* based on the example of a single gene and a ~15% decrease in methylation from a single experiment. The embryos at the examined stages are already differentiated and the tissue complexity can be a confounding factor in such analyses. The authors should attempt a more robust approach for detecting DNA methylation differences such as RRBS and WGBS and demonstrate more substantial links between the absence of Fam60a and a decrease in DNA methylation. To minimize the influence of tissue complexity at the desired embryonic stages, the experiment could be performed on tissues most affected by the loss of Fam60a such as heart or intestine. Also, to substantiate the link between Tet1 and *Fam60*, the authors should explore the developmental dynamics of the two proteins through ChIP-seq profiling in mouse embryos.

*Reviewer #3:*

The authors have attempted to address the biological role of Fam60a. They show that the protein shows widespread expression during early mouse development; the expression becomes progressively restricted. The authors have next generated *Fam60a* knockout and show that the deletion of this gene leads to embryonic lethality after E10.5. In order to understand the role of Fam60a on the molecular level, the authors have identified Fam60a protein interactors – this surprisingly revealed Sin3a complex. Fam60a was further shown to interact with gene promoters by ChIP-seq with the knockout leading both to gene activation and repression. While there is no obvious effect on AcH3k9 of the target genes, the link is made to Tet1 and DNA methylation through evolutionary conservation, this is followed by rather preliminary finding of Fam60a having a negative regulatory role on Tet1 activity.

Overall, I have a number of major comments:

1) The characterisation of the generated mouse knockout requires an evidence at the Fam60a transcript or protein level.

2) Is Sin3a complex affected in the absence of *Fam60a*?

3) The "evolutionary analysis of conservation" – this is certainly an interesting observation, however it should not be overstated as it lacks the rigor of the standard conservation and co-evolution analysis.

4) ChIP-seq analysis: +/-3kb is too broad for the promoter region. The small overlap between Fam60a targets (ChIP-seq) and the genes down regulated in the KO suggests that the majority of the downregulated genes are probably due to secondary effects.

5) My main comment – the link between Tet1 and Fam60a is too preliminary to make any claims. I would definitely expect the authors to answer the type of questions they are raising in their Discussion to warrant publication of their study.

[Editors’ note: what now follows is the decision letter after the authors submitted for further consideration.]

Thank you for submitting your article "Fam60a is a component of the Sin3a complex and regulates DNA demethylation at a subset of gene promoters" for consideration by *eLife*. Your article has been reviewed by two peer reviewers, and the evaluation has been overseen by a Reviewing Editor and Marianne Bronner as the Senior Editor. The reviewers have opted to remain anonymous.

The reviewers have discussed the reviews with one another and the Reviewing Editor has drafted this decision to help you prepare a revised submission.

The authors have carried out further experiments on line with the reviewers' suggestions. However a number of points, mainly involving restructuring the manuscript (points 1 and 2), together with some further data analysis if possible (points 3 and 4) and data presentation (point 5), need to be addressed:

1) The regulation of DNA methylation should not be overemphasised since it did not completely convince the reviewers. We suggest this be toned down in its presentation with a title change along the lines of: "Loss of *Fam60*, a Sin3 subunit results in XXX and is associated with hypomethylation on XXX promoters".

2) The manuscript would read better if it started with the biochemistry experiments i.e. largely confirming Streubel et al. findings. This could then be followed by the evolutionary comparisons, mice phenotypes, and genomics/transcriptomics data.

3) The DNA methylation data should be analysed more carefully. If possible, the authors need to show heatmaps with DNA methylation levels over Fam60-bound promoters in the wild type and the knockout. This will immediately clarify to what extent Fam60 is associated with changes in DNA methylation.

4) The average changes in DNA methylation in hyper- and hypo-methylated DMRs need to be included in the paper. This is important so that the reader can get an idea of what fraction of methylation is really changing upon the loss of *Fam60*.

5) The authors should try not to copy and paste the plots directly from the programs that generated them into the manuscript. Most of the plots in this paper are copy-pasted screenshots. This type of visualisation is substandard.

*Reviewer #1:*

The authors have performed additional experiments to try and strengthen the hypotheses presented in the initial submission. They have invested a significant effort into improving this manuscript, especially when taking into account that they moved their lab from Osaka to Kobe during this period.

The initial descriptions of the *Fam60a^-/-^* phenotypes and evolutionary studies have improved significantly. Also, additional validations associated with their genome-wide observations were helpful. Nevertheless, I still find the link between *Fam60* and DNA demethylation unsatisfying and I believe that more useful information could have been extracted from base-resolution bisulfite sequencing data (i.e. extent of mC change in the detected DMRs, heatmaps of mC over *Fam60*-bound promoters in wt and *Fam60a^-/-^* etc.). Also, some essential QC details such as non-conversion rates are missing.

Essentially, based on the presented data, I cannot find obvious links between *Fam60* and DNA demethylation. The reasons for this are following:

It is stated in the title and throughout the manuscript that "*Fam60* regulates DNA demethylation on a subset of promoters". Based on experiments presented in Figure 6, the reader is led to believe that *Fam60*, through an undefined mechanism, somehow inhibits Tet1 activity. Logically, this would then mean that a subset of promoters that is normally hypermethylated, would become hypomethylated in the *Fam60a^-/-^* condition. Nevertheless, the DMR analysis does not support this scenario as most hypomethylated DMRs appear associated with intronic regions. Also, this "subset" of promoters is not identified in the manuscript and remains limited to a handful of examples that display modest DNA methylation differences, (Figure 7), that could just be attributed to pleiotropic effects.

In total, I am supportive of this manuscript, however, I would urge the authors to:

1) Change the title and remove any sentences/paragraphs that claim links between *Fam60a^-/-^* and DNA demethylation.

2) To discuss their findings better in the context of a recently published study (Streubel et al., 2017), where Fam60a was described as a Sin3 complex component. While the Streubel et al. study is indeed briefly mentioned in the Discussion, more credit should be given there.

*Reviewer #2:*

The manuscript entitled "Fam60a is a component of the mSin3a-Hdac transcriptional corepressor complex and inhibits Tet-mediated DNA demethylation" shows that Fam60a modulates DNA methylation, possibly by modulating Tet activity. Because a manuscript published in EMBO J in 2017 describes the link between Fam60a, mSin3a-Hdac and Tet1, the primary finding in this manuscript is the knockout mouse phenotype, which is incomplete, and the mechanistic link with Tet1, which remains ambiguous. This paper is not referenced.

The authors examine expression of *Fam60a*, but the best example is in the intestinal crypt; even so, this does not link with the story. A knockout shows that embryos die in mid-gestation, and have general organ hypoplasia as well as cardiac outflow tract defects. A *Fam60a::Venus* BAC transgenic mouse was used to rescue the phenotype. Under several different IP conditions, they identified several important interacting partners, including mSin3A and HDAC1, but not Tet1. Tet1 was found as a partner in the competing paper. ChIP-seq revealed that Fam60a is recruited to promoter regions of genes to regulate gene expression, these data are compared with RNASeq to identify putative targets. Methylation differs at these target loci, suggesting that demethylation increases in the absence of *Fam60a*. The authors show convincing evidence that 5-hydroxymethylation, a product of Tet activity, increases when Fam60a is absent in transfected NIH3T3 cells, yet do not provide direct evidence for how Fam60a might be doing so. An evolutionary comparison suggests that Fam60a evolved at around the same time as Tet1. Even so, how differential methylation by Tet is modulated by Fam60a remains obscure, and the link to Tet1 is superficial. This manuscript does not extend or clarify the findings presented in the other manuscript.

Moreover, the EMBO J manuscript shows that the complex binds to the H3K4me3 mark. In this regard, did H3K4me3 change at the targets identified? AcH3K9, a target of HDAC, did not differ.

---

## [Author Response]

[Editors’ note: the author responses to the first round of peer review follow.]

During revision of our manuscript, a related paper titled “Fam60a defines a variant Sin3a-Hdac complex in embryonic stem cells required for self-renewal” was published by Streubel et al., 2017. The data in this paper overlap in part and are consistent with our results. Streubel et al. thus also show that Fam60a is a component of the Sin3a-Hdac complex. However, the two studies are substantially different in that Streubel et al. examine the role of Fam60a in ES cells, whereas we have characterized the phenotype of Fam60a knockout mice. Furthermore, we now also present evidence that Fam60a controls gene expression at least in part by regulating the DNA methylation status of a subset of gene promoters. Our paper thus still reports many new findings, and we believe that it is worthy of publication in *eLife*. We now refer to the study of Streubel et al. in our revised manuscript (Discussion, second paragraph) and therefore include it in the list of references.

Reviewer #1:The manuscript entitled "Fam60a is a component of the mSin3a-Hdac transcriptional corepressor complex and inhibits Tet-mediated DNA demethylation" shows that Fam60a modulates DNA methylation, possibly by modulating Tet activity. The impact of this finding is that no function has been previously associated with Fam60a, and its possible association with methylation in stem cells is significant.Having previously found differential expression of Fam60a in ESCs and differentiated cells, the authors examined expression throughout development and in adults to find that expression correlated with somatic stem cells in the intestinal crypt. They developed a Fam60a-CreERT2 transgene to examine lineage expression in crypt cells, and found expression in the villi that migrated up to the tip. They then developed a knockout to find that embryos die in mid-gestation, and have general organ hypoplasia as well as cardiac outflow tract defects. They developed a Fam60a::Venus BAC transgenic mouse to identify interacting proteins using IP-mass spec. Interestingly, mSin3A and HDAC1 were found as strong interacting partners. They carried out ChIP-seq to find that Fam60a is recruited to promoter regions of genes to regulate gene expression, and compared these data with RNASeq. The authors show convincing evidence that 5-hydroxymethylation, a product of Tet activity, increases when Fam60a is absent in transfected NIH3T3 cells, yet do not provide direct evidence for how Fam60a might be doing so. A further concern is that upon comparing mutant and wild type embryos for differential methylation at putative Fam60a target genes, only one gene appears to be differentially methylated. Therefore, how differential methylation by Tet is modulated by Fam60a remains obscure.The manuscript reads as if it is two different papers. One story touches on the role of Fam60a in development and intestinal stem cells, and the second assesses its biochemical function. This is somewhat of a problem for genes for which no function is known, but I feel that the manuscript would be more appealing to a wider audience if it could be tightened up somewhat to focus on the role of Fam60a in stem cells and methylation. The story should also be linked again in the Discussion.

We now provide genome-wide methylation data (new Figure 8 and Figure 8—figure supplements 1 and 2) and focus on the role of Fam60a in DNA methylation. Our results suggest that Fam60a controls gene expression at least in part by regulating DNA methylation at a subset of gene promoters. We have also modified the Discussion section to address the possible link between the role of Fam60a in development and its biochemical function (Discussion, first paragraph).

The embryo expression at E9.5 would be very difficult for anyone who is not an expert to interpret, and could be placed in supplementary, focusing on intestinal expression. The lineage tracing experiment in intestinal villi is very nice, but loses relevance in Figure 1. Further, arrows pointing to key regions would help to guide the reader. It is not clear if the expression data correlate with the heart defects seen in the knockout, or if the KO mice have intestinal defects.

We have reorganized these data according to the reviewer’s suggestions. The revised Figure 1 now focuses on intestinal expression of Fam60a and includes the lineage tracing experiment for intestinal villi. The remaining expression data have been moved to Figure 1—figure supplement 1. The *Fam60a* KO mice manifest both heart and intestinal defects. Abnormal gut looping in the mutant embryos is now shown in the new Figure 2—figure supplement 3. We also now show *Fam60a* expression in the developing heart (new Figure 2G). *Fam60a* is expressed predominantly in the outflow tract and right ventricle of the heart at E13.5, consistent with our observation that the right ventricle is hypoplastic in KO mice. We modified the text of the Results section accordingly (subsection “Developmental defects in *Fam60a* mutant mice”, first paragraph).

Some data are confusing and do not support the general claims of the authors. For example, ChIP-Seq identified a large number of putative gene targets, yet differential methylation was found at only one – Adhfe1. Without definition of these genes, interpretation of any biology is difficult. Sin3a-Hdac would normally alter AcH3K9, yet this mark did not change at Fam60a target genes in WT and mutant embryos. Some discussion of these unexpected data would be warranted.

We have now included data on genome-wide DNA methylation in WT and *Fam60a* KO embryos, showing that many promoter regions (including that of *Adhfe1*) are hypomethylated in the mutant embryos (new Figure 8 and Figure 8—figure supplements 1 and 2). As pointed out by the reviewer, the amount of AcH3K9 at Fam60a target genes was not altered in the mutant embryos. We have now performed a similar analysis for Fam60a target genes in control and *Fam60a*^–/–^ ES cells and found that the level of AcH3K9 at these genes was significantly increased in the mutant cells (Figure 4—figure supplement 2C). These data are now described in the Results section (subsection “Fam60a is recruited to promoter regions and regulates gene expression”, last paragraph).

Reviewer #2:The manuscript by Nabeshima et al. explores the role of Fam60a, a protein highly expressed in ESCs. The first part of the manuscript describes the phenotype of the mouse Fam60 mutant and the protein interactions of Fam60 including the Sin3A complex components. This is followed by genomic analyses of Fam60 binding and the transcriptome analyses of the mutant. The authors then attempt to build a link between Fam60 and Tet1 in cultured cells and mouse embryos. In my opinion, the majority of the genomic data appears very preliminary and the presented findings frequently do not support the drawn conclusions. The links between Fam60 action and Tet1 are especially weak and built mostly on conjecture. I also found the quality of the figures (especially the figures dealing with ChIP-seq and RNA-seq data) substandard while many other figures are not properly labelled and thus confusing. The biochemical data are of some interest due to the broad spectrum of Sin3 functions, unfortunately however, those experiments were not followed up on. My comments and suggestions for improvement are outlined below.What is the reason for including the MA plot in Figure 4A? It comes with very little/no explanation and if its only purpose is to show that Fam60 loss results in bidirectional gene changes, then it is redundant with the B panel.

We have now omitted this plot in the revised manuscript.

Did the authors assess whether the upregulated and downregulated genes are enriched in any specific GO categories? Some gene network analyses would also be very beneficial since there are currently no strong links between the mouse phenotypes and the RNA-seq data.

As suggested by the reviewer, we performed gene ontology analysis, with the results being shown in new Figure 4—figure supplement 1. We found that genes related to nutrient responses are up-regulated and those related to lipid biosynthesis are down-regulated in the mutant embryos. These findings are now described in the text of the Results section (subsection “Fam60a is recruited to promoter regions and regulates gene expression”, first paragraph).

Why are there no error bars in Figure 4C? I assume that this data corresponds to the average of the three RNA-seq experiments.

The RNA-seq data are indeed from three independent experiments, and error bars have now been added to the figure.

I find it very problematic that the Fam60 ChIP-seq experiment has no sequenced input controls (at least I couldn't find any). This can result in suboptimal peak calling and biased downstream analyses. For example, the observed enrichment of Fam60 peaks around the TSS could be partly due to the GC content sequencing bias.

The input DNA has been sequenced by NGS, which is included in the data deposited to DNA Data Bank of Japan (DDBJ) with the accession numbers DRA004842.

What is the reason for the double peak in Figure 5A?

Sin3a and H3K4me3 also show two peaks in previous studies by other researchers (for example, see Bowman et al., 2014). The two peaks seem to represent the regions immediately upstream and downstream of the transcription start site.

What does "+/- 3Kb" means in Figure 5C? Were the peaks extended for 3Kb in each direction before the overlaps were performed?

The designation “ ± 3 kb” means the region between –3 kb and +3 kb relative to the transcription start site, as has now been clarified in the legend.

Subsection “Fam60a inhibits Tet1 activity in cultured cells”: A Table is not an adequate tool for evolutionary comparisons. To properly address this issue, the authors should construct phylogenetic trees for DNMT proteins, Tet proteins, Sin3 and Fam60a. Also, how did the authors decide which organisms to include in this comparison? I would suggest including basal metazoans (such as Cnidaria and Ctenophora that are known to have DNA methylation) as well as cephalochordates and tunicates. A few insects known to employ DNA methylation (such as ants and wasps) can also be included.

We have now constructed a phylogenetic tree for Fam60 proteins and provided a more detailed analysis of the phylogeny of Fam60, Dnmt, Tet, and Sin3 proteins and their relation to the absence or presence of DNA methylation (new Figure 6A and B).

Figure 6: The data presented in Figure 6 is not very convincing. The effect of Fam60 on Tet1 expression is not very robust and the statistical significance of the presented differences appears to be low. Also, I find the system that the authors use highly artificial. A better experiment would be to generate Fam60a^-/-^ cells (for example ESCs that express both proteins) and perform a Tet1 ChIP-seq to assess whether the loss of Fam60 results in increased binding of Tet1. This experiment could also be complemented with base-resolution hmC profiling.

As suggested by the reviewer, we generated *Fam60a*^–/–^ ES cells, which retained pluripotency markers, and performed a Tet1 ChIP assay as well as an AcH3K9 ChIP assay (new Figure 4—figure supplement 2). The level of AcH3K9 at the promoter regions of three Fam60a target genes was significantly increased in *Fam60a*^–/–^ ES cells compared with control cells (Figure 4—figure supplement 2C). However, the level of Tet1 at the promoters of these genes was not affected by the absence of Fam60a (Figure 4—figure supplement 2D). The recruitment of Tet1 to these promoter regions thus appeared to be independent of Fam60a. Given that Fam60a appeared to attenuate Tet1 function in transfected cells (Figure 6C-F), Fam60a may regulate Tet1 by inhibiting its enzymatic activity rather than by preventing its recruitment to gene promoters. We have now addressed this issue in the text (Discussion, last paragraph).

Subsection “Aberrant promoter hypomethylation in Fam60a^–/–^ mouse embryos” and Figure 7: It is very difficult to make any definite conclusions regarding the relationships between Tet1 and Fam60 based on the example of a single gene and a ~15% decrease in methylation from a single experiment. The embryos at the examined stages are already differentiated and the tissue complexity can be a confounding factor in such analyses. The authors should attempt a more robust approach for detecting DNA methylation differences such as RRBS and WGBS and demonstrate more substantial links between the absence of Fam60 and a decrease in DNA methylation. To minimize the influence of tissue complexity at the desired embryonic stages, the experiment could be performed on tissues most affected by the loss of Fam60 such as heart or intestine. Also, to substantiate the link between Tet1 and Fam60, the authors should explore the developmental dynamics of the two proteins through ChIP-seq profiling in mouse embryos.

As suggested by the reviewer, we examined DNA methylation more comprehensively in WT and *Fam60a*^–/–^ embryos. The results are now shown in the new Figure 8 and Figure 8—figure supplements 1 and 2 and are described in the text (subsection “Differentially methylated regions in the genome of *Fam60a^–/–^* embryos”). The body and CpG shore regions of all genes rather than the whole genome were examined by bisulfite sequencing, given that this approach is equally meaningful and more effective. There was no apparent difference in the global methylation level between WT and mutant embryos, with ~45% of CpGs being methylated in both. However, we detected 7245 differentially methylated regions (DMRs), 3049 of which were hypermethylated and 4196 hypomethylated in the *Fam60a* mutant embryos. Among the top 500 DMRs of each type, 83 (17%) hypermethylated and 254 (51%) hypomethylated regions corresponded to Fam60a binding regions detected by ChIP-seq, with the hypomethylated DMRs thus being enriched in Fam60a binding sites. As expected, *Adhfe1* was included in the top 500 hypomethylated DMRs (Figure 8—figure supplement 2).

Reviewer #3:The authors have attempted to address the biological role of Fam60a. They show that the protein shows widespread expression during early mouse development; the expression becomes progressively restricted. The authors have next generated Fam60a knockout and show that the deletion of this gene leads to embryonic lethality after E10.5. In order to understand the role of Fam60a on the molecular level, the authors have identified Fam60a protein interactors – this surprisingly revealed Sin3a complex. Fam60 was further shown to interact with gene promoters by ChIP-Seq with the knockout leading both to gene activation and repression. While there is no obvious effect on AcH3k9 of the target gene, the link is made to Tet1 and DNA methylation through evolutionary conservation, this is followed by rather preliminary finding of Fam60a having a negative regulatory role on Tet1 activity.Overall, I have a number of major comments:1) The characterisation of the generated mouse knockout requires an evidence at the Fam60a transcript or protein level.

We now provide RT-qPCR and immunoblot data (Figure 2—figure supplement 1C and D) showing the loss of Fam60a mRNA and protein in the mutant embryos. These results are also briefly mentioned in the text (subsection “Developmental defects in *Fam60a* mutant mice”, first paragraph).

2) Is Sin3a complex affected in the absence of Fam60a?

We examined proteins co-immunoprecipitated with Sin3a from extracts of *Fam60a*^–/–^ and control ES cells (new Figure 3—figure supplement 3). The formation of the Sin3a-Hdac complex did not appear to be affected by the absence of Fam60a. These results are now described in the text (subsection “*Fam60a* interacts with components of the Sin3a-Hdac complex”, last paragraph).

3) The "evolutionary analysis of conservation" – this is certainly an interesting observation, however it should not be overstated as it lacks the rigor of the standard conservation and co-evolution analysis.

We have now constructed a phylogenetic tree for Fam60 proteins and provided a more detailed analysis of the phylogeny of Fam60, Dnmt, Tet, and Sin3 proteins and their relation to the absence or presence of DNA methylation (new Figure 6A and B). We have also toned down the description of this issue to avoid overstatement (subsection “Association of Fam60a with DNA methylation and Tet”, first paragraph).

4) ChIP-Seq analysis: +/-3kb is too broad for the promoter region. The small overlap between Fam60a targets (ChIP-Seq) and the genes down regulated in the KO suggests that the majority of the downregulated genes are probably due to secondary effects.

As shown in Figure 4A, only 26% (45/172) of down-regulated genes in the mutant mice contain a Fam60a binding site, suggesting that attenuation of the expression of most of the down-regulated genes is due to a secondary effect of Fam60a loss. We have now addressed this point in the Results section (subsection “Fam60a is recruited to promoter regions and regulates gene expression”, last paragraph). Promoter regions as defined as ± 1 kb relative to the transcription start site constitute only 1.1% of the genome (new Figure 5—figure supplement 1B) but contain 13.4% or 16.8% of Fam60a binding sites in two independent ChIP-seq experiments (Figure 5B and Figure 5—figure supplement 1B), indicating that Fam60a binding sites are indeed enriched at promoter regions.

5) My main comment – the link between Tet1 and Fam60a is too preliminary to make any claims. I would definitely expect the authors to answer the type of questions they are raising in their Discussion to warrant publication of their study.

To address the link between Tet1 and Fam60a, we examined the association of Tet1 and AcH3K9 with the promoters of three Fam60a target genes in *Fam60a*^–/–^ and control ES cells (new Figure 4—figure supplement 2). The level of AcH3K9 was significantly increased at each of these promoter regions in the *Fam60a*^–/–^ ES cells (Figure 4—figure supplement 2C). In contrast, the level of Tet1 associated with these promoters was not affected by the absence of Fam60a (Figure 4—figure supplement 2D). The recruitment of Tet1 to these promoter regions thus appears to be independent of Fam60a. Given that forced expression of Fam60a attenuated Tet1 function in NIH3T3 cells (Figure 6C–F), Fam60a may regulate Tet1 by inhibiting its enzymatic activity rather than by preventing its recruitment to gene promoters. We have now addressed this issue in the text of the revised manuscript (Discussion, last paragraph). We have also changed the title of our paper to remove reference to Tet1.

[Editors' note: the author responses to the re-review follow.]

The authors have carried out further experiments on line with the reviewers' suggestions. However a number of points, mainly involving restructuring the manuscript (points 1 and 2), together with some further data analysis if possible (points 3 and 4) and data presentation (point 5), need to be addressed:1) The regulation of DNA methylation should not be overemphasised since it did not completely convince the reviewers. We suggest this be toned down in its presentation with a title change along the lines of: "Loss of Fam60, a Sin3 subunit results in XXX and is associated with hypomethylation on XXX promoters".

We have changed the title according to the suggestion. The new title is “Loss of Fam60a, a component of the Sin3a complex, results in embryonic lethality and is associated with aberrant methylation at a subset of gene promoters.” We used “aberrant methylation” instead of hypomethylation because a significant portion of DMRs is hypermethylated.

2) The manuscript would read better if it started with the biochemistry experiments i.e. largely confirming Streubel et al. findings. This could then be followed by the evolutionary comparisons, mice phenotypes, and genomics/transcriptomics data.

We have re-structured the manuscript according to the editors’ suggestion. Thus the paper starts with the biochemistry experiments, which is followed by the evolutionary comparisons, mice phenotypes, and genomics/transcriptomics data.

3) The DNA methylation data should be analysed more carefully. If possible, the authors need to show heatmaps with DNA methylation levels over Fam60-bound promoters in the wild type and the knockout. This will immediately clarify to what extent Fam60 is associated with changes in DNA methylation.

As suggested by the editors, we now show heatmaps with DNA methylation levels over Fam60-bound promoters in the wild type and the knockout (Figure 8—figure supplement 2). Hypomethylation was commonly observed at the Fam60a-binding regions. While a number of DMRs were clearly detected, there was no obvious difference in this type of presentation between the wild-type and *Fam60a*^–/–^ embryos (Figure 8—figure supplement 2). This is probably because only a subset of gene promoters was affected in the mutant (Figure 8A), and because the changes of DNA methylation between the wild-type and the mutant are small (Figure 8—figure supplement 1, Figure 8—figure supplement 3).

4) The average changes in DNA methylation in hyper- and hypo-methylated DMRs need to be included in the paper. This is important so that the reader can get an idea of what fraction of methylation is really changing upon the loss of Fam60.

We now show the average changes in DNA methylation in hyper- and hypo-methylated DMRs in Figure 8—figure supplement 3. Among 7245 DMRs detected, the average changes of DNA methylation were 11.87% and 10.99% for hyper- and hypomethylated DMRs, respectively (subsection "Differentially methylated regions in the genome of *Fam60a^–/–^* embryos”, second paragraph).

5) The authors should try not to copy and paste the plots directly from the programs that generated them into the manuscript. Most of the plots in this paper are copy-pasted screenshots. This type of visualisation is substandard.

We have improved the quality of all figures such as by removing copy-pasted screenshots.